Genomic Prediction

# Genetic prediction with ARG-powered linear algebra

Hanbin Lee (iD),[1,*,†] Nathaniel S. Pope (iD),[2,†] Jerome Kelleher (iD),[3] Gregor Gorjanc (iD),[4] Peter L. Ralph (iD)[2]

[1]Department of Statistics, University of Michigan, Ann Arbor, MI 48109, United States
[2]Institute of Ecology & Evolution, University of Oregon, Eugene, OR 97405, United States
[3]Big Data Institute, Li Ka Shing Centre for Health Information and Discovery, University of Oxford, Oxford OX3 7LF, United Kingdom
[4]The Roslin Institute and Royal (Dick) School of Veterinary Studies, University of Edinburgh, Edinburgh EH25 9RG, United Kingdom

*Corresponding author: Department of Statistics, University of Michigan, Ann Arbor, MI 48109, United States. Email: hblee@umich.edu
[†]These authors contributed equally to this work.

Ancestral recombination graphs (ARGs) are an attractive means for quantitative genetic analysis of complex traits because they encode the realized genetic relatedness between a sample of individuals in the presence of genetic drift, recombination, and mutation. Data structures for efficiently storing ARGs can also be used to rapidly process millions of genomes, and are thus promising for fitting linear mixed models to large phenotype and genome datasets. Here, we study the problems of variance component estimation and prediction of genetic values with ARGs, by describing a generative model of complex traits with additive effects on an ARG, and then developing algorithms that use the ARG to solve these problems efficiently on biobank-scale datasets. We observe nearly linear scaling of runtime with sample size, which is achieved by using the succinct tree sequence representation of the ARG for implicit matrix-vector products, along with modern randomized linear algebra algorithms. We estimate variance components using restricted maximum likelihood, which we find performs substantially better than the Haseman–Elston method. In simulation tests, both variance component estimation and prediction of genetic values (using the best linear unbiased predictor) perform nearly as well with inferred ARGs as with true ARGs. We also discuss interpretations of the variance component estimates as mutational variance and additive genetic variance. We provide an implementation of the algorithms as a Python package tslmm, which leverages the tree sequence library tskit.

Keywords: ancestral recombination graph; genetic prediction; polygenic score; linear mixed model; genomic prediction

## Introduction

The ancestral recombination graph (ARG) encodes the genetic relationships between a set of recombining genomes (Hudson 1990; Griffiths and Marjoram 1997; Wong et al. 2024). Recent methods are capable of reconstructing ARGs from population genomic data (reviewed in Brandt et al. 2024; Lewanski et al. 2024), providing a window into the history of the genomes from which it is possible to identify signatures of natural selection as well as demographic events that happened in the past (e.g. Stern et al. 2021; Pope et al. 2023). Some methods scale to hundreds of thousands of samples (Kelleher et al. 2019; Zhang et al. 2023), allowing applications to biobanks and large modern animal or plant breeding datasets. There has long been interest in using ARGs to analyze complex (quantitative) traits (e.g. Larribe et al. 2002; Minichiello and Durbin 2006; see Selle et al. 2021; Link et al. 2023 for more citations). Following the recent advances in ARG inference (Rasmussen et al. 2014; Kelleher et al. 2019; Speidel et al. 2019; Zhang et al. 2023; Gunnarsson et al. 2024; Deng et al. 2025), several groups have used inferred ARGs to identify genome regions harboring genetic variation for complex traits (Link et al. 2023; Zhang et al. 2023; Gunnarsson et al. 2024; Rebollo et al. 2025; Zhu J et al. 2025).

Many methods for the quantitative genetic analysis of complex traits use linear mixed models (LMMs), which use prior genetic information through a genetic relatedness matrix (GRM). A GRM can be obtained from pedigree, phylogeny, genotypes, or the underlying ARG (reviewed in Lehmann et al. 2026). Work thus far has used the ARG to define a GRM in terms of the total amount of shared ancestral material ("area of shared edges," as described later), and so is the expected value of the genotype GRM (VanRaden 2008; Speed and Balding 2015) given the ARG structure and the infinite-sites mutation model. This has been referred to as a "ARG-GRM," "eGRM," or "branch GRM." Since there are in principle many ways to obtain a GRM from an ARG, in this paper we use the term "branch GRM" (Lehmann et al. 2026).

Recent developments show promising results for the use of ARGs in quantitative genetics. Zhang et al. (2023) proposed a Monte-Carlo algorithm for estimating the branch GRM and used it to estimate heritability and variant associations. They showed that the branch GRM can improve heritability estimation and, by using simulated variants on the ARG, complement traditional imputation approaches to detect signals from ungenotyped variants. Link et al. (2023) summarized the aggregate signal of variants within a local window by computing the branch GRM from local trees spanning the window. When applied to a Native Hawaiian cohort, the method successfully found signals that were overlooked by conventional GWAS due to low variant density and lack of insufficient imputation panels. Rebollo et al. (2025) compared the use of pedigree, genotype, and branch GRMs for prediction of genetic values in a rice dataset with two sub-species, and found that using the branch GRM outperforms other GRMs.

Rebollo et al. (2025) also backsolved the predicted genetic values to variant associations and demonstrated how these combine into haplotype values in local trees.

Zhang et al. (2023), Link et al. (2023), and Rebollo et al. (2025) used ARGs to compute GRM for downstream analysis, but did not take full advantage of the computational potential of inferred ARGs. They all used existing software (e.g. GCTA—Yang et al. 2011 or ASReml—Butler et al. 2023) and substituted the conventional genotype GRM with their branch GRM version to fit the LMM. This approach is computationally challenging: with $N$ individuals storing a GRM scales with $N^2$ while many downstream computations with a GRM scale with $N^3$. Another challenge in computing the GRM is the large number of distinct local trees as one moves along the genome. However, the ARG itself can be seen as a sparse representation of both the local trees and the genotypes themselves (Ané and Sanderson 2005; Wong et al. 2024), and can be used to compute a broad range of statistics of both, orders of magnitude faster than variant-by-variant or tree-by-tree approaches (Ralph et al. 2020; Lehmann et al. 2026; Zhu J et al. 2025).

Zhu J et al. (2025) was the first paper to utilize this computational potential of the ARG to fit an LMM using the branch GRM. They devised a randomized Haseman–Elston (HE) method, called ARG-RHE, that combines randomized linear algebra with an ARG matrix-vector multiplication algorithm that recycles information shared by adjacent local trees to minimize redundant computations. Their algorithm allows estimating local variance components with biobank-scale datasets to conduct GWAS, which is very difficult with traditional methods. However, room remains for improvement: their underlying algorithms compute quantities based on branch lengths of local trees using a Monte-Carlo approximation with randomly generated mutations, and thus incur a tradeoff between accuracy and computational resources. Furthermore, the HE method, as a moment-based estimator (Haseman and Elston 1972), is fast but has lower statistical precision compared to the restricted maximum likelihood (REML), which utilizes the full likelihood (Patterson and Thompson 1971).

In this paper, we study the problem of *genetic prediction*: predicting the genetic values of a certain set of individuals based on the phenotypic values of other individuals and some knowledge about relatedness between them (Henderson 1975; Lynch and Walsh 1998; Wray et al. 2019; Mrode and Pocrnic 2023). This shares many features with the problem of trait mapping (e.g. Strandén and Garrick 2009; Fan et al. 2022; Zhang et al. 2023), and many approaches to both use the same linear mixed model:

$$\mathbf{y} = \mathbf{Xb} + \mathbf{g} + \boldsymbol{\varepsilon}, \tag{1}$$

where $\mathbf{y}$ is the vector of phenotypic values (observed and unobserved), $\mathbf{X}$ is a matrix of covariates and $\mathbf{b}$ are the corresponding coefficients, $\mathbf{g}$ is a vector of the genetic values, and $\boldsymbol{\varepsilon}$ is a vector of uncorrelated, Gaussian, "environmental" deviations. Prediction can be done by finding the conditional mean of the unobserved entries of $\mathbf{g}$ given the observed entries. Information about relatedness enters by assuming that the genetic values are similar for closely related individuals: with the covariance between $\mathbf{g}_i$ and $\mathbf{g}_j$ proportional to the "relatedness" between individuals $i$ and $j$. The Gaussian assumption reduces the main problem—estimating $\mathbf{b}$ and $\mathbf{g}$ given the observed data—to linear algebra involving $\mathbf{X}$ and the relatedness matrix. For instance, if $\mathbf{R}$ is the relatedness matrix and $\mathbf{I}$ is the identity, then the covariance matrix of $\mathbf{y}$ is $\tau^2 \mathbf{R} + \sigma_\varepsilon^2 \mathbf{I}$, where $\tau^2$ and $\sigma_\varepsilon^2$ are unknown values controlling the relative contributions of genetics and unmodeled

noise, respectively (and thus, heritability). The most challenging step is estimation of the *variance components* $\tau^2$ and $\sigma_\varepsilon^2$. Given these, it is fairly straightforward to estimate $\mathbf{b}$ and $\mathbf{g}$.

Oversimplifying somewhat, methods differ in the computational algorithms and the choice of relatedness; here we use the ARG to both provide the measure of relatedness and to speed up computation. To this end, we first describe in some detail the generative model of complex traits on an ARG, which leads to the ARG-based linear mixed model (ARG-LMM) (Zhang et al. 2023). Next, we describe how to fit the ARG-LMM to very large datasets, obtaining estimates of variance components and predicted genetic values ("best linear unbiased predictions," or BLUPs). Computation is a major obstacle for large datasets, so here we focus on using the ARG and randomized linear algebra for efficient inference. The methods are implemented in the open-source Python package `tslmm` available at https://github.com/hanbin973/tslmm, which builds on the tree sequence `tskit` library, to take advantage of the project's extensive code base, documentation, unit testing, and stable API (Jeffery et al. 2026).

## The ARG linear mixed model (ARG-LMM)

The linear mixed model framework applied to ARGs was first presented in Zhang et al. (2023) where they assume that Var($\mathbf{g}$), the covariance matrix of the genetic value $\mathbf{g}$ in equation (1), is the branch GRM. By explicitly expressing the genetic value in terms of ARG components, beginning from the standard genotype-based representation $\mathbf{g} = \mathbf{G}\boldsymbol{\beta}$, we next demonstrate how this assumption follows from first principles. The derivation is most naturally justified by assuming neutrality, since then mutational effects are independent of genealogical relationships. Suppose that for the $i$th individual we have a phenotypic value $\mathbf{y}_i$, a set of covariates $\mathbf{X}_{i1}, \dots, \mathbf{X}_{iK}$, and genotypes $\mathbf{G}_{i1}, \dots \mathbf{G}_{iP}$. The equation that is commonly used to define the linear mixed model assumes that the phenotypic value of individual $i$, is equal to the sum of fixed effects, $\sum_j \mathbf{X}_{ij}\mathbf{b}_j$, its genetic value, $\mathbf{g}_i = \sum_p \mathbf{G}_{ip}\boldsymbol{\beta}_p$, and its environmental/non-genetic deviation $\boldsymbol{\varepsilon}_i$; in matrix form this is:

$$\mathbf{y} = \mathbf{Xb} + \mathbf{G}\boldsymbol{\beta} + \boldsymbol{\varepsilon}. \tag{2}$$

Here $\mathbf{b}$ and $\boldsymbol{\beta}$ are coefficients for the non-genetic covariates and genotypes, respectively. However, the features of the sample ARG are not visible in this equation. Hence, we seek an alternative expression of equation (2) that explicitly contains ARG-related terms.

We work with the "infinitesimal" model of phenotype evolution on the ARG, that is used implicitly or explicitly in a variety of papers; see Lehmann et al. (2026) for a recent review. The model can be thought of as the extension to a recombining genome of the "Brownian motion" model of phenotype evolution on phylogenies used by Felsenstein (1973) and Thompson (1975), and recently discussed by Schraiber et al. (2024). The model can also be thought of as the specialization of the classical infinitesimal model (Fisher 1918; Barton et al. 2017) to the situation where we know not only the pedigree of individuals but also which segments of genome they have inherited from parental genomes within the pedigree or from ancestral genomes beyond the pedigree. This model assumes that the net effects of any mutations on a phenotype are additive, and so the genetic covariance in phenotypic values between any pair of individuals is proportional to the total amount of shared ancestral opportunity for mutation.

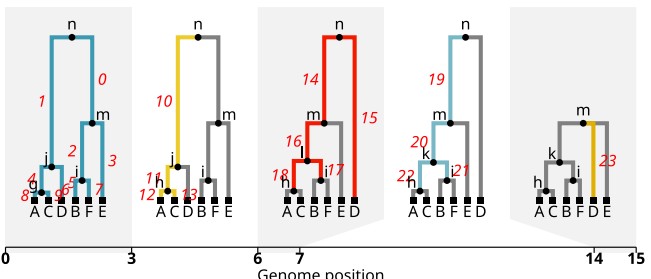

**Fig. 1.** A short tree sequence with sub-edges labeled: moving left to right, each new sub-edge is colored and given a label; sub-edges that remain the same from the previous tree are grey. There are 23 total sub-edges but fewer edges: for instance, the edge from (parent) $n$ to (child) $m$ is split into three sub-edges, labeled 0, 14, and 19.

Here, we propose an explicit phenotype model that induces the branch GRM. To make the model concrete, an ARG $\mathcal{T}$ is defined using collections of nodes $\mathcal{N}$ and edges $\mathcal{E}$. Each node $n$ represents a (possibly ancestral) chromosome, and so has a (birth) time $t_n$, measured in time ago (i.e. time before the present). A special set of nodes are the *samples*, for which we have complete genome information. Each edge represents a segment of genome spanning $[a_e, b_e]$ that was inherited by a child node $c_e$ from a parent node $p_e$. (The "child" and "parent" terms refer to relationships in the graph, and so these may be separated by more than one generation.) Along the segment $[a_e, b_e]$, the subtree descending from $c_e$ may change due to further recombinations. Suppose that we split the genomic segment for each edge into disjoint pieces, which we call *sub-edges*, so that there are no recombinations in the ARG below each sub-edge. An example is illustrated in Fig. 1. The induced graph structure on sub-edges is a *multitree* (Furnas and Zacks 1994), which has a number of useful properties (Nowbandegani et al. 2023; DeHaas et al. 2024). For instance, values cached at nodes are updated once per sub-edge in the algorithms in Ralph et al. (2020) and Zhu J et al. (2025).

With the above assumptions and taking an infinite-sites mutation model (Kimura 1969; Watterson 1975), any mutations that occur along a sub-edge $e$ are inherited by all nodes below the sub-edge. Subsequently, the sample nodes below a sub-edge remain constant along its whole span. Thus, we can define the "sub-edge dosage" design matrix $\mathbf{Z}$, with rows indexed by individuals and columns indexed by sub-edges, so that $\mathbf{Z}_{ie}$ is equal to the number of haplotypes the individual $i$ inherited from sub-edge $e$ (for diploids, $\mathbf{Z}_{ie} \in \{0, 1, 2\}$). This allows us to express the genetic value $\mathbf{g} = \mathbf{G}\boldsymbol{\beta}$ as a linear function of sub-edge dosages $\mathbf{Z} \in \mathbb{R}^{N \times E}$ and sub-edge effects $\mathbf{u} \in \mathbb{R}^E$, which turns equation (2) into

$$\mathbf{y} = \mathbf{Xb} + \mathbf{Zu} + \boldsymbol{\varepsilon}. \tag{3}$$

Here the sub-edge effects $\mathbf{u}$ represent the net effects of any mutations that occur on each sub-edge; further details of this conversion can be found in Model formulation and assumptions. We assume that $\boldsymbol{\varepsilon}$ are independent, have mean 0 and variance $\sigma_\varepsilon^2$. What about $\mathbf{u}$? The total opportunity for mutation of an sub-edge is equal to the per base-pair mutation rate multiplied by its "area," $A_e = (b_e - a_e)(t_{p_e} - t_{c_e})$, which is its genomic span multiplied by its time length. Thus, an "infinitesimal" model would assume that each $\mathbf{u}_e$ have mean zero and variance $\tau^2 A_e$; see Model formulation and assumptions for more careful discussion of this point. This implies that the covariance matrix of $\mathbf{Zu}$ is $\tau^2 \mathbf{B} = \tau^2 \mathbf{Z} \boldsymbol{\Sigma}_A \mathbf{Z}^T$, where $\boldsymbol{\Sigma}_A$ is the diagonal matrix with areas $A_e$ on the diagonal. $\mathbf{B}$ is the

branch GRM using the definition in Lehmann et al. (2026). $\tau^2$ is the variance of mutational effects $\mathbf{u}$ per generation and base-pair.

In summary, the generative model assumes that:

1) genetic values are additive on the ARG;
2) environmental effects are independent with variance $\sigma_\varepsilon^2$;
3) and the net effects of each sub-edge are independent,
4) have mean zero,
5) and variance proportional to sub-edge area,

An implication of this model is the "ARG linear mixed model" (ARG-LMM): let $\mathbf{Z}$ be the individual/sub-edge design matrix defined in the text, and $\boldsymbol{\Sigma}_A$ the diagonal matrix with the areas of each sub-edge. Then the covariance of the observed phenotypic values $\mathbf{y} = \mathbf{Xb} + \mathbf{Zu} + \boldsymbol{\varepsilon}$ is:

$$\tau^2 \mathbf{Z} \boldsymbol{\Sigma}_A \mathbf{Z}^T + \sigma_\varepsilon^2 \mathbf{I},$$

and the covariance of the genetic values $\mathbf{g} = \mathbf{Zu}$ is $\tau^2 \mathbf{B}$, where $\tau^2$ is the variance of mutational effects per generation, per base-pair, and $\mathbf{B}$ is the branch GRM,

$$\mathbf{B} = \mathbf{Z} \boldsymbol{\Sigma}_A \mathbf{Z}^T.$$

Note that under this model the distribution of sub-edge effects does not depend on the frequency $p$ of the sub-edge (i.e. the number of samples that would inherit any SNPs on that edge). It is common (including in the ARG context by Fan et al. 2022; Zhang et al. 2023) to normalize these effects by $(p(1 - p))^\alpha$ for some value of $\alpha$. It is straightforward to incorporate this into the methods, but is hard to deduce it from the generative model for the following reason: frequency is determined after the mutations materialize, hence including it as part of the generative model ahead of time is conceptually counterintuitive. Thus, we leave the precise model formulation incorporating frequency information to future work.

## Assumptions

An advantage to explicitly formulating the generative model is that it makes the assumptions visible, and suggests ways forward if the assumptions seem unlikely to produce good results. Note, however, that this is *a* generative model that leads to the LMM that we use below (and is used elsewhere; Fan et al. 2022; Zhang et al. 2023; Rebollo et al. 2025); other assumptions may arrive at the same model. Here we discuss issues conceptually; see Model formulation and assumptions for a more rigorous formulation.

While equation (3) implies an additive model without epistasis or dominance, the model can also be thought of as a first-order approximation in the presence of such non-additive effects. Independence of sub-edges is violated when there are multiple causal mutations on related sub-edges, that is one sub-edge is ancestral to another, at the same site. For instance, if the same allele appears independently on different sub-edges, then both alleles are assumed to have the same effect under the additive model. As quantified in Model formulation and assumptions, the correlations thus induced are small.

The assumption that effects have mean zero could easily be violated. For instance, we expect most mutations in coding regions to be deleterious, so mutations in DNA repair pathways might have a mean positive effect on cancer risk. However, this assumption turns out to be relatively unimportant. As discussed in Model formulation and assumptions, mean effects are absorbed into the intercept that is typically included in $\mathbf{X}$ as long as all individuals are taken at roughly the same time.

The assumption that variance is proportional to area is perhaps the strongest. One way to justify it is as follows: suppose that mutations occur at rate $\mu$ per base-pair, and the effects of these mutations are additive, with mean zero and variance $\sigma^2$. If mutations cannot obscure the effect of each other (i.e. a continuum-of-alleles model, as in Kimura and Crow 1964), then the variance of the total effect of the mutations on a given sub-edge $e$ is $\tau^2 A_e$ with $\tau^2 = \mu\sigma^2$. If furthermore, $\mu$ is large and $\sigma$ is small, then by the central limit theorem, the net effect of those mutations is Gaussian, as in Barton et al. (2017). See also Schraiber and Landis (2015) and Koch (2019), who discuss a similar neutral model while averaging over the ARG. Our calculation takes the point of view that (a) the net effect of any mutations occurring along an sub-edge is unknown (and random); but also (b) the mutations themselves are unobserved. The latter point might seem odd, because we observe genotypes more directly than ARGs—shouldn't the variance of the effects of an sub-edge be proportional to the number of mutations actually on it, rather than the *potential* for mutations? Perhaps, but we find the point of view useful in a few ways. First, it is unlikely that all causal alleles are genotyped, or that all genotyped polymorphisms are causal, and so estimates of the branch GRM have the potential to better reflect a shared genetic basis for the phenotype. Indeed, Zhang et al. (2023) and Link et al. (2023) identified putative ungenotyped causal alleles following this point of view. Second, the framework makes it natural to distinguish properties of the underlying generative process (such as the mutational variance) from descriptors of the studied samples (such as the additive genetic variance). Finally, the methods we describe can be easily adapted by replacing sub-edge area with the number of mutations on the sub-edge.

A further implication of the last assumption is that causal mutations fall uniformly along the genome. In practice, we might have a prior idea of where these might fall, thus leading to a heterogeneous rate. This could be accounted for by changing the genomic coordinate system to reflect the opportunity for causal mutations. Similarly, if the phenotype is under selection, then it is commonly assumed that causal mutations are unlikely to be common. This might be accounted for by rescaling time, so that causal mutations occur uniformly along the rescaled time axis. These are discussed in more detail in Model formulation and assumptions.

## Efficient GRM-vector multiplication, and ARG-LMM simulation

The key to computational efficiency in subsequent sections is the ability to use the tree sequence as, effectively, a sparse matrix representation with which we can efficiently perform GRM-vector products. In Lehmann et al. (2026), we showed that this computation amounts to "adding up values down an ARG." To see why this is, consider the problem of multiplying the GRM by $\mathbf{1}_N$, the constant vector of 1's of length $N$. The (uncentered) GRM is $\mathbf{Z}\Sigma_A\mathbf{Z}^T$, so we want to compute $\mathbf{v} = \mathbf{Z}\Sigma_A\mathbf{Z}^T\mathbf{1}_N$. Since $\mathbf{Z}_{ie} = 1$ if sample $i$ is below sub-edge $e$, and $[\Sigma_A]_{ee}$ is the area of sub-edge $e$, $\mathbf{v}$ is obtained by assigning to each sub-edge a weight which is the product of the number of samples below the sub-edge multiplied by its area, and then assigning to $\mathbf{v}_i$ the sum of weights of all sub-edges above sample $i$. (Note that this "down" step is preceded by an "up" step to obtain the number of samples below each sub-edge.) However, a naïve tree-by-tree implementation of this description would not take advantage of the substantial structure shared across many trees.

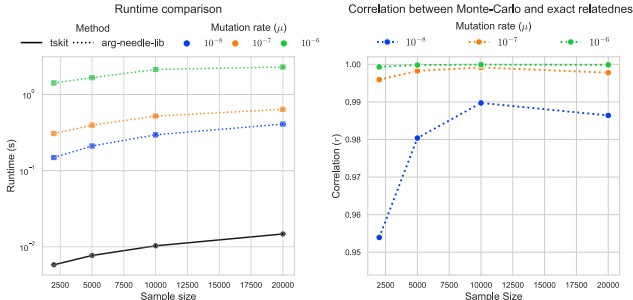

**Fig. 2.** Timing and accuracy comparison of ARG-based GRM-vector multiplication algorithms implemented in `tskit` and `arg-needle-lib`. We repeated 10 replicates of tree sequence and mutation simulations with effective population size of $N_e = 10^5$, recombination rate of $10^{-8}$, and sequence length of $10^6$ bp. The left panel shows the average runtime of two methods across varying mutation rate. The right panel shows the average Pearson correlation between Monte-Carlo (`arg-needle-lib`) and exact (`tskit`) GRM-vector products results. Note that $\mu = 10^{-6}$ is the default setting of `arg-needle-lib`.

"Adding up values down an ARG" is precisely how the generative model of the previous section works: the genetic value of a sample is equal to the sum of the effects of all sub-edges ancestral to the sample, so it turns out that an efficient algorithm for simulation from the generative model has the same shape as the matrix-vector product algorithm. We describe an algorithm to do just this in Correctness of Algorithm T, which we find useful for clarifying the dependency structures within the ARG, and why using the full ARG structure is substantially more efficient than tree-by-tree methods.

Although sub-edges are central here, they are not explicitly formed in the algorithms we present in this paper, because creating and storing them can incur a significant computational burden. Instead, we work with the original edges stored in the tree sequences to implement what is algorithmically equivalent to operations on sub-edges.

A closely related algorithm was presented in Zhu J et al. (2025) and implemented in `arg-needle-lib`. `tskit` performs GRM-vector multiplication by doing the "up" and "down" steps mentioned above in one pass. On the other hand, `arg-needle-lib` splits the process into left and right multiplications of the simulated genotype matrix. More crucially, `arg-needle-lib`'s algorithm is approximate: it uses the number of randomly generated mutations as a proxy for branch area, and so the runtime grows with increasing mutation rate. To demonstrate these points, we've compared `tskit` and `arg-needle-lib`'s GRM-vector product algorithms. The benchmark was conducted on genomes of length $10^6$ bp simulated from `msprime` under a constant size panmictic population ($N_e = 10^5$) with a recombination rate of $10^{-8}$ over 10 replicate simulations. `arg-needle-lib`'s results depend on the value of $\mu$, the rate at which the generated mutations are placed on the ARG: Fig. 2b shows that for these ARGs, $\mu$ close to $10^{-6}$ is needed to obtain nearly exact values (which are provided by `tskit`). Nonetheless, `tskit` is one to two orders of magnitude faster than `arg-needle-lib` across this range of values of $\mu$ (Fig. 2a).

## Variance component estimation

Now, we turn to estimation of the variance components, which we do using a modified version the average information restricted maximum likelihood (AI-REML) algorithm (Gilmour et al. 1995).

This section derives the REML algorithm as applied to the ARG-LMM model, and describes the computational methods we use to apply it to large-scale problems in `tslmm`.

As a model of phenotypes $\mathbf{y}$, ARG-LMM is a multivariate Gaussian distribution. (Although the derivation is based on a Gaussian model, the resulting REML estimator is robust to distributional error provided the GRM is correct; see **Discussion**.) Hence, we'd like to obtain estimates of $\tau^2$ and $\sigma_\varepsilon^2$ in the ARG-LMM model, which is, concisely:

$$\mathbf{y} \mid \mathbf{X}, \mathbf{b}, \tau^2, \sigma_\varepsilon^2 \sim \text{Normal}(\mathbf{Xb}, \mathbf{V}) \quad \text{with} \quad \mathbf{V} = \tau^2\mathbf{B} + \sigma_\varepsilon^2\mathbf{I}, \tag{4}$$

where $\mathbf{y}$ is the vector of phenotypic values, $\mathbf{B} = \mathbf{Z}\Sigma_A\mathbf{Z}^T$ as before, $\mathbf{X}$ is an $N \times K$ matrix of covariates, $N$ is the sample size (the length of $\mathbf{y}$), $K$ is the number of covariates, and $\mathbf{b}$ is the vector of coefficients. Rather than directly finding the maximum likelihood estimates using the whole model, it turns out to be advantageous to instead find $\tau^2$ and $\sigma_\varepsilon^2$ that maximize the likelihood of the portion of $\mathbf{y}$ that is orthogonal to the columns of $\mathbf{X}$. This leads to the restricted maximum likelihood (REML) objective (Patterson and Thompson 1971), which is, up to a constant factor:

$$\begin{aligned}\ell(\mathbf{y}, \mathbf{X}; \tau^2, \sigma_\varepsilon^2) = {} & (N - K)\log(2\pi) + \log\det(\mathbf{V}) \\ & + \log\det(\mathbf{X}^T\mathbf{V}^{-1}\mathbf{X}) + \mathbf{y}^T\mathbf{Py},\end{aligned} \tag{5}$$

where $\mathbf{P} = \mathbf{V}^{-1} - \mathbf{V}^{-1}\mathbf{X}(\mathbf{X}^T\mathbf{V}^{-1}\mathbf{X})^{-1}\mathbf{X}^T\mathbf{V}^{-1}$. If certain assumptions are met, REML is unbiased and statistically the most efficient estimator of variance components (Patterson and Thompson 1971; Harville 1977; Westfall 1987; Zhou 2017).

Thus, our goal is to find $\tau^2$ and $\sigma_\varepsilon^2$ that maximize (5). To explain how this is done, we turn to general-purpose optimization theory. Consider an unconstrained optimization problem for a twice continuously differentiable function $f : \mathbb{R}^p \to \mathbb{R}$, finding $\boldsymbol{\theta}^* = \text{argmin}_{\boldsymbol{\theta} \in \mathbb{R}^p} f(\boldsymbol{\theta})$. (We will have $\theta = (\tau^2, \sigma_\varepsilon^2)$, and so $f(\tau^2, \sigma_\varepsilon^2) = \ell(\mathbf{y}, \mathbf{X}; \tau^2, \sigma_\varepsilon^2)$.) The Newton-Raphson method seeks to find an approximate solution by iterating:

$$\boldsymbol{\theta}_{n+1} = \boldsymbol{\theta}_n - \left[\nabla_{\boldsymbol{\theta}}^2 f(\boldsymbol{\theta}_n)\right]^{-1} \cdot \nabla_{\boldsymbol{\theta}} f(\boldsymbol{\theta}_n), \tag{6}$$

where $\nabla_{\boldsymbol{\theta}} f$ is the gradient of $f$ and $\nabla_{\boldsymbol{\theta}}^2 f$ is the Hessian.

Fisher scoring is a variant of the Newton-Raphson method when the optimization target includes realized random variables. Suppose $\mathbf{x}$ is a random variable that is generated from a distribution parameterized by $\boldsymbol{\theta}$. The optimization target (e.g. the negative log-likelihood) $\ell(x, \theta)$ is a function of both $\boldsymbol{\theta}$ and $x$ where $x$ is a realization of $\mathbf{x}$, and so we now want to find $\boldsymbol{\theta}^* = \text{argmin}_{\boldsymbol{\theta} \in \mathbb{R}^p} \ell(x, \boldsymbol{\theta})$. The corresponding Newton-Raphson update is:

$$\boldsymbol{\theta}_{n+1} = \boldsymbol{\theta}_n - \left[\nabla_{\boldsymbol{\theta}}^2 \ell(x, \boldsymbol{\theta}_n)\right]^{-1} \nabla_{\boldsymbol{\theta}} \ell(x, \boldsymbol{\theta}_n). \tag{7}$$

Fisher scoring replaces $\nabla_{\boldsymbol{\theta}}^2 \ell(x, \boldsymbol{\theta}_n)$ by its expectation:

$$\boldsymbol{\theta}_{n+1} = \boldsymbol{\theta}_n - \mathbb{E}_{\boldsymbol{\theta}_n}\left[\nabla_{\boldsymbol{\theta}}^2 \ell(\mathbf{x}, \boldsymbol{\theta}_n)\right]^{-1} \nabla_{\boldsymbol{\theta}} \ell(x, \boldsymbol{\theta}_n), \tag{8}$$

where the random variable $\mathbf{x}$ sampled at $\boldsymbol{\theta} = \boldsymbol{\theta}_n$ replaces the realized observation $x$. The expectation $\mathbb{E}_{\boldsymbol{\theta}}\left[\nabla_{\boldsymbol{\theta}}^2 \ell(\mathbf{x}, \boldsymbol{\theta})\right]$ is called the *Fisher information*.

Average information is the average of these two quantities:

$$\text{AI}(x, \theta) = \frac{1}{2}\left(\nabla_{\boldsymbol{\theta}}^2 \ell(x, \boldsymbol{\theta}) + \mathbb{E}_{\boldsymbol{\theta}}\left[\nabla_{\boldsymbol{\theta}}^2 \ell(\mathbf{x}, \boldsymbol{\theta})\right]\right), \tag{9}$$

and the update in iteration is:

$$\boldsymbol{\theta}_{n+1} = \boldsymbol{\theta}_n - \text{AI}(x, \boldsymbol{\theta}_n)^{-1} \nabla_{\boldsymbol{\theta}} \ell(x, \boldsymbol{\theta}_n). \tag{10}$$

All three methods—Newton–Raphson, Fisher scoring, and average information—are second-order optimization methods. Roughly speaking, second-order methods locally approximate the optimization target as a quadratic function by utilizing curvature information encoded in the second derivative (the Hessian). While Newton-Raphson uses the Hessian directly, Fisher scoring uses the Hessian's average over the data distribution at the current step, and average information uses the average of the two. As quadratic approximation is more accurate than linear approximation used by first-order methods, second-order methods generally require fewer iterations and converge more tightly than first-order methods, although quadratic methods are often computationally more expensive than first-order methods because computing the Hessian or related quantities can be very demanding (Boyd and Vandenberghe 2004). To this end, we develop an efficient average information REML (AI-REML) routine for ARG-LMM with a minimal additional overhead beyond a stochastic estimate of the gradient. First, we derive expressions for the AI-REML updates (which have useful features not found in the general case; Gilmour et al. 1995), and then we discuss how to compute these efficiently.

## Average information residual maximum likelihood (AI-REML)

To find the average information update (equation 10), we need the first and second derivatives of the REML objective (equation 5) with respect to $\tau^2$ and $\sigma_\varepsilon^2$. (Readers not interested in the computations may skip to the punchline at the end of this section.) Since for any invertible-matrix-valued function $A(x)$, $\partial_x \log\det A(x) = \text{tr}(A(x)^{-1}\partial_x A(x))$, then after a little algebra using the cyclic property of the trace, the first derivatives are:

$$\begin{aligned}\tfrac{\partial\ell}{\partial\sigma_\varepsilon^2} &= \text{tr}(\mathbf{P}(\nabla_{\sigma_\varepsilon^2}\mathbf{V})) - \mathbf{y}^T\mathbf{P}(\nabla_{\sigma_\varepsilon^2}\mathbf{V})\mathbf{Py} \\ &= \text{tr}(\mathbf{P}) - \mathbf{y}^T\mathbf{PPy} \\ \tfrac{\partial\ell}{\partial\tau^2} &= \text{tr}(\mathbf{P}(\nabla_{\tau^2}\mathbf{V})) - \mathbf{y}^T\mathbf{P}(\nabla_{\tau^2}\mathbf{V})\mathbf{Py} \\ &= \text{tr}(\mathbf{PB}) - \mathbf{y}^T\mathbf{PBPy},\end{aligned} \tag{11}$$

because $\nabla_{\sigma_\varepsilon^2}\mathbf{V} = \mathbf{I}$ and $\nabla_{\tau^2}\mathbf{V} = \mathbf{B}$. Similarly, the second derivatives are:

$$\begin{aligned}\tfrac{\partial\ell}{\partial\sigma_\varepsilon^2\partial\sigma_\varepsilon^2} &= -2\mathbf{y}^T\mathbf{P}(\nabla_{\sigma_\varepsilon^2}\mathbf{V})\mathbf{P}(\nabla_{\sigma_\varepsilon^2}\mathbf{V})\mathbf{Py} + \text{tr}(\mathbf{P}(\nabla_{\sigma_\varepsilon^2}\mathbf{V})\mathbf{P}(\nabla_{\sigma_\varepsilon^2}\mathbf{V})) \\ \tfrac{\partial\ell}{\partial\sigma_\varepsilon^2\partial\tau^2} &= -2\mathbf{y}^T\mathbf{P}(\nabla_{\sigma_\varepsilon^2}\mathbf{V})\mathbf{P}(\nabla_{\tau^2}\mathbf{V})\mathbf{Py} + \text{tr}(\mathbf{P}(\nabla_{\sigma_\varepsilon^2}\mathbf{V})\mathbf{P}(\nabla_{\tau^2}\mathbf{V})) \\ \tfrac{\partial\ell}{\partial\tau^2\partial\tau^2} &= -2\mathbf{y}^T\mathbf{P}(\nabla_{\tau^2}\mathbf{V})\mathbf{P}(\nabla_{\tau^2}\mathbf{V})\mathbf{Py} + \text{tr}(\mathbf{P}(\nabla_{\tau^2}\mathbf{V})\mathbf{P}(\nabla_{\tau^2}\mathbf{V})).\end{aligned} \tag{12}$$

The expectations of the second derivatives, averaging over values of $\mathbf{y}$, are:

$$\begin{aligned}\mathbb{E}\left[\tfrac{\partial\ell}{\partial\sigma_\varepsilon^2\partial\sigma_\varepsilon^2}\right] &= -\text{tr}(\mathbf{P}(\nabla_{\sigma_\varepsilon^2}\mathbf{V})\mathbf{P}(\nabla_{\sigma_\varepsilon^2}\mathbf{V})) \\ \mathbb{E}\left[\tfrac{\partial\ell}{\partial\sigma_\varepsilon^2\partial\tau^2}\right] &= -\text{tr}(\mathbf{P}(\nabla_{\sigma_\varepsilon^2}\mathbf{V})\mathbf{P}(\nabla_{\tau^2}\mathbf{V})) \\ \mathbb{E}\left[\tfrac{\partial\ell}{\partial\tau^2\partial\tau^2}\right] &= -\text{tr}(\mathbf{P}(\nabla_{\tau^2}\mathbf{V})\mathbf{P}(\nabla_{\tau^2}\mathbf{V})).\end{aligned} \tag{13}$$

Finally, averaging the Hessian (12) and the Fisher information (13) yields the average information:

$$\begin{aligned}\text{AI}(\sigma_\varepsilon^2, \sigma_\varepsilon^2) &= -\mathbf{y}^T\mathbf{P}(\nabla_{\sigma_\varepsilon^2}\mathbf{V})\mathbf{P}(\nabla_{\sigma_\varepsilon^2}\mathbf{V})\mathbf{Py} \\ &= -\mathbf{y}^T\mathbf{PPPy} \\ \text{AI}(\sigma_\varepsilon^2, \tau^2) &= -\mathbf{y}^T\mathbf{P}(\nabla_{\sigma_\varepsilon^2}\mathbf{V})\mathbf{P}(\nabla_{\tau^2}\mathbf{V})\mathbf{Py} \\ &= -\mathbf{y}^T\mathbf{PPBPy} \\ \text{AI}(\tau^2, \tau^2) &= -\mathbf{y}^T\mathbf{P}(\nabla_{\tau^2}\mathbf{V})\mathbf{P}(\nabla_{\tau^2}\mathbf{V})\mathbf{Py} \\ &= -\mathbf{y}^T\mathbf{PBPBPy},\end{aligned} \tag{14}$$

where, notably, the computationally-expensive traces have canceled. This cancelation is the biggest appeal of AI-REML in general, a fact that holds even for multiple variance components as long as the overall covariance matrix **V** is a linear combination of the parameters (Gilmour et al. 1995; Zhu S and Wathen 2018). Hence, our terms for the AI-REML update (10) are:

$$
\begin{aligned}
\nabla_{\boldsymbol{\theta}}\ell &= \begin{bmatrix} \mathrm{tr}(\mathbf{P}) - \mathbf{y}^T\mathbf{P}\mathbf{P}\mathbf{y} \\ \mathrm{tr}(\mathbf{P}\mathbf{B}) - \mathbf{y}^T\mathbf{P}\mathbf{B}\mathbf{P}\mathbf{y} \end{bmatrix} \\
\mathrm{AI}(\ell) &= \begin{bmatrix} \mathrm{AI}(\sigma_\varepsilon^2, \sigma_\varepsilon^2) & \mathrm{AI}(\sigma_\varepsilon^2, \tau^2) \\ \mathrm{AI}(\sigma_\varepsilon^2, \tau^2) & \mathrm{AI}(\tau^2, \tau^2) \end{bmatrix}.
\end{aligned}
\tag{15}
$$

Our next goal is to describe how to compute these values efficiently.

## Efficient computation with AI-REML

To apply AI-REML, we will need to begin with an initial guess at $\tau^2$ and $\sigma_\varepsilon^2$, then repeatedly iterate the updates (10) using the expressions in (15). Since these expressions involve products of $N \times N$ matrices, and the update itself requires inversion of such matrices, doing this explicitly is impractical for large $N$. Instead, we now describe how to carry out all required computations using implicit matrix-vector multiplication with the ARG. The basic tool we have for this is the algorithm presented in Lehmann et al. (2026), which allows us to multiply a vector by **B** with nearly linear complexity in the number of individuals; this is implemented in tskit as genetic_relatedness_vector.

The most expensive single operation we need to perform is multiplication by **P**. First, note that to multiply by **P** it suffices to multiply by $\mathbf{V}^{-1}$; since **X** is only $N \times K$, multiplication by **X** is done explicitly. To multiply by $\mathbf{V}^{-1}$, we use the conjugate gradient method (Hestenes and Stiefel 1952), an iterative algorithm which multiplies **V** to a vector multiple times to obtain $\mathbf{V}^{-1}\mathbf{x}$. Since $\mathbf{V} = \tau^2\mathbf{B} + \sigma_\varepsilon^2\mathbf{I}$, multiplication by **V** is no harder than multiplication by **B**. Multiplication by **P** lets us compute all terms in the AI matrix, and the non-trace terms in the gradient. Some further simplifications are possible: for instance, to compute $\mathbf{y}^T\mathbf{P}\mathbf{P}\mathbf{y}$, we first estimate the fixed effects by $\widehat{\mathbf{b}} = (\mathbf{X}^T\mathbf{V}^{-1}\mathbf{X})^{-1}\mathbf{X}^T\mathbf{V}^{-1}\mathbf{y}$. Then, we compute **Py** by $\mathbf{P}\mathbf{y} = \mathbf{V}^{-1}(\mathbf{y} - \mathbf{X}\widehat{\mathbf{b}})$, and $\mathbf{y}^T\mathbf{P}\mathbf{P}\mathbf{y} = \|\mathbf{P}\mathbf{y}\|^2$.

The next ingredient we need for the gradient is the ability to compute traces, which turns out to be the most computationally intensive step. For instance, $\mathrm{tr}(\mathbf{P})$ in $\partial\ell/\partial\sigma_\varepsilon^2$ is:

$$
\begin{aligned}
\mathrm{tr}(\mathbf{P}) &= \mathrm{tr}(\mathbf{V}^{-1}) - \mathrm{tr}(\mathbf{V}^{-1}\mathbf{X}(\mathbf{X}^T\mathbf{V}^{-1}\mathbf{X})^{-1}\mathbf{X}^T\mathbf{V}^{-1}) \\
&= \mathrm{tr}(\mathbf{V}^{-1}) - \mathrm{tr}((\mathbf{X}^T\mathbf{V}^{-1}\mathbf{X})^{-1}\mathbf{X}^T\mathbf{V}^{-1}\mathbf{V}^{-1}\mathbf{X}),
\end{aligned}
\tag{16}
$$

First note that $\mathbf{X}^T\mathbf{V}^{-1}\mathbf{X}$ is only a $K \times K$ matrix, so can be inverted explicitly, and that furthermore the second trace term is a trace of a $K \times K$ matrix, which can be computed easily. The difficult part of computing $\mathrm{tr}(\mathbf{P})$ is computation of $\mathrm{tr}(\mathbf{V}^{-1})$. We obtain an unbiased estimate of each trace with the XTrace algorithm, which uses a fixed number of matrix-vector multiplications against random vectors (Epperly et al. 2024). In our default setting, we use 50 **P** multiplications for the estimation of each of the traces (see Table 1), which is conservative in our experience.

Combining these tools—implicit matrix-vector multiplication by **B**, conjugate gradient for inverses, and XTrace for traces—we have highly efficient ways to compute AI-REML updates. There are a few more details for the full algorithm. First, we use a randomized Nyström preconditioner (Frangella et al. 2021) to speed up the conjugate gradient algorithm to perform multiplication by $\mathbf{V}^{-1}$. This preconditioner uses a randomized singular value decomposition that depends on the number of test vectors used.

**Table 1.** The default hyperparameter settings of tslmm.

| Hyperparameter name | Default value |
| --- | --- |
| Conjugate gradient convergence threshold | $10^{-5}$ |
| Number of test vectors for gradient estimation | 50 |
| Rank of the randomized Nyström preconditioner | 500 |
| Number of test vectors for randomized HE method | 50 |
| Number of threads in matrix-vector multiplication | 1 |
| REML convergence criterion | 0.05 |
| Minimum REML steps before termination | 15 |

Similarly, trace estimation with the XTrace algorithm (Epperly et al. 2024) depends on a chosen number of test vectors. Default values for these and other parameters are listed in Table 1. Our implementation of AI-REML method is iterative and stochastic, and so a stopping criterion is needed. When the parameter estimates change by less than 5%, tslmm continues for an additional 15 steps and then terminates, returning the average value of the last 15 iterations.

How efficient is it? The matrix-vector algorithm for multiplication with **B** performs an $\mathcal{O}(N)$ setup step, then one update for each node above each added or removed edge, and so has computational complexity $\mathcal{O}(N + E\log N)$, where $N$ is the number of individuals and $E$ is the number of edges (Lehmann et al. 2026). In practice, runtime of variance component estimation is highly dependent on the number of matrix-vector multiplications, which we minimized through the use of modern randomized algorithms. Most matrix-vector operations are performed on many vectors, and so are parallelized across the number of available threads (our benchmarking results shown below, however, used a single thread). Furthermore, note that while the gradient is stochastic, the average information matrix is not; hence this AI-REML optimization routine is more stable than "fully-stochastic" second-order methods.

## Initialization by Haseman–Elston

The AI-REML algorithm is iterative, so it needs a starting estimate. We initialize the variance component parameters using a randomized Haseman–Elston (HE) method. This is attractive because it can be simply and directly obtained without resorting to optimization methods, so is a relatively cheap starting point for the more accurate iterative method. For a derivation, see Wu and Sankararaman (2018), but briefly the initial values of $\tau^2$ and $\sigma_\varepsilon^2$ are obtained by solving:

$$
\begin{bmatrix} \mathrm{tr}(\mathbf{B}^2) & \mathrm{tr}(\mathbf{B}) \\ \mathrm{tr}(\mathbf{B}) & \mathrm{tr}(\mathbf{I}) \end{bmatrix} \begin{bmatrix} \tau^2 \\ \sigma_\varepsilon^2 \end{bmatrix} = \begin{bmatrix} \mathbf{y}^T\mathbf{B}\mathbf{y} \\ \mathbf{y}^T\mathbf{y} \end{bmatrix},
\tag{17}
$$

in which the traces are computed by XTrace (Epperly et al. 2024) with 50 test vectors as above. Computing $\mathrm{tr}(\mathbf{B}^2)$ and $\mathrm{tr}(\mathbf{B})$ is far cheaper than a single AI-REML iteration because multiplication with **P** requires multiple rounds of **B** multiplication. Then, solving equation (17) is immediate. Therefore, initialization by the randomized HE method only occupies a small fraction of the overall AI-REML routine.

## Simulations for evaluation of methods

Next, we produced a collection of simulations to benchmark the methods. Briefly, we used msprime (Kelleher et al. 2016; Baumdicker et al. 2022) for a coalescent simulation of an exponentially growing panmictic population. Looking back through time (following the coalescent simulation), the population starts at

100,000 individuals today, and declines at a rate of 1% per generation. Each individual has a genome that was either 1 Mbp ($10^6$ bp) or 100 Mbp ($10^8$ bp) (depending on the situation, below) and experienced a recombination rate of $10^{-8}$ per base-pair per generation. From the resulting population we sampled varying numbers of diploid individuals.

For real data, we do not have access to the true ARGs. Hence, for some tests we used `msprime` to simulate mutations on the tree sequences with rate $10^{-8}$ per base-pair per generation to obtain haplotypes, which were used in `tsinfer` (Kelleher et al. 2019) and `tsdate` (Wohns et al. 2022; Pope et al. 2026) (with the "variational gamma" algorithm) to obtain inferred and dated tree sequences.

To simulate phenotypes, we first simulated genetic values using the algorithm described in apx:algoT, then added independent environmental noise, as in the ARG-LMM model. (In all benchmarks, phenotypes were generated from the true simulated tree sequence, not inferred tree sequences.) To make the variance due to genetic and non-genetic components roughly comparable, we divided the genetic values by the square root of $4 \sum_T s_T t_{\text{root},T}$, where the sum is over local trees and $t_{\text{root},T}$ is the time of the root of a local tree $T$. This is because the genetic variance of a single haploid genome is proportional to $\sum_T s_T t_{\text{root},T}$, the total area from the single genome up to the root along the entire genome, and so the genetic variance of a diploid is bounded above by four times this quantity. The final phenotypic value for each individual was obtained by multiplying their genetic value by $\tau$ and adding an independent Gaussian environmental deviation with mean 0 and variance $\sigma_\varepsilon^2$.

## Computational scaling

To assess the scalability of the methods, we simulated phenotypes described above on ARGs, using $\tau^2 = 1$ and $\sigma_\varepsilon^2 = 1$, and estimated the variance components with `tslmm`. We did this five times on each simulated tree sequence to obtain an average runtime.

The results show that `tslmm` conforms to the predicted scaling. Variance component estimation for $10^5$ individuals and a one-chromosome genome took only a few hours on a single thread. Runtimes for `tslmm` on simulated datasets for 100 Mbp diploid genomes are shown in Fig. 3, as a function of increasing ARG size. Although matrix-vector multiplication using the algorithm of Lehmann et al. (2026) is $\mathcal{O}(N + E \log(N))$, the number of multiplications required for conjugate gradient could also scale with $N$ or $E$, resulting in different computational complexity. However, the lines in Fig. 3 suggest that this is not the case: empirical runtimes for variance component estimation are fit well by an $\mathcal{O}(N + E \log(N))$ model.

We also compared runtime to a modern, fast method for variance component estimation from genotypes, `BOLT-LMM`. This method's runtime scales with $\mathcal{O}(N^{1.5}M)$, where $M$ is the number of variants (Loh et al. 2015). Since $M$ increases with sample size, we expect `BOLT-LMM`'s runtime to increase more rapidly than `tslmm`'s. Figure 4 confirms this: `tslmm` is faster than `BOLT-LMM` for sample sizes above about $2 \times 10^4$, and the gap widens with an increasing number of samples. For this test, all variants produced from the `msprime` simulation was passed to `BOLT-LMM`, so the number of variants $M$ also increased with the number of individuals $N$ in the benchmark; filtering of variants would reduce `BOLT-LMM`'s runtime. Also note that `BOLT-LMM`'s variance component estimation is a subroutine of its' larger GWAS pipeline. Therefore, one should not interpret `tslmm`, which does not perform GWAS, as a replacement for `BOLT-LMM`. Instead, the finding suggests that one could borrow `tslmm`'s machinery to speed up

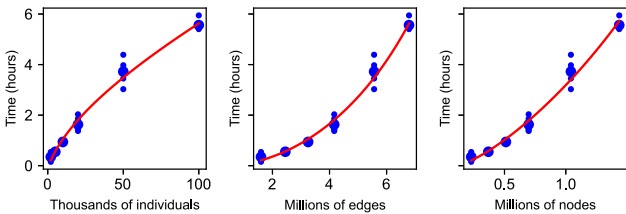

**Fig. 3.** Measured runtimes (on a single thread of Xeon Gold 6,140 CPU) for variance component estimation on 30 simulations of a 100 Mbp genome with a recombination rate of $10^{-8}$: five independent simulations at each of six different sample sizes of diploid individuals. Since runtime is expected to scale with the number of edges, we show the same results against number of diploid individuals (left), total number of edges (middle), and number of nodes (right). Larger dots show the mean across replicates, and lines show results from the linear model "runtime = 87 (# individuals/$10^3$) + 365 (# edges/$10^6$) log(# individuals/$10^3$) − 185." The linear model was fit by least squares, and displayed red lines are obtained by using the average value of the unspecified variable across simulated scenarios with the given value of the specified variable (for instance, in the first plot, the line shows predicted runtime at the number of individuals shown on the x-axis and the average number of edges seen across the simulations having that many individuals).

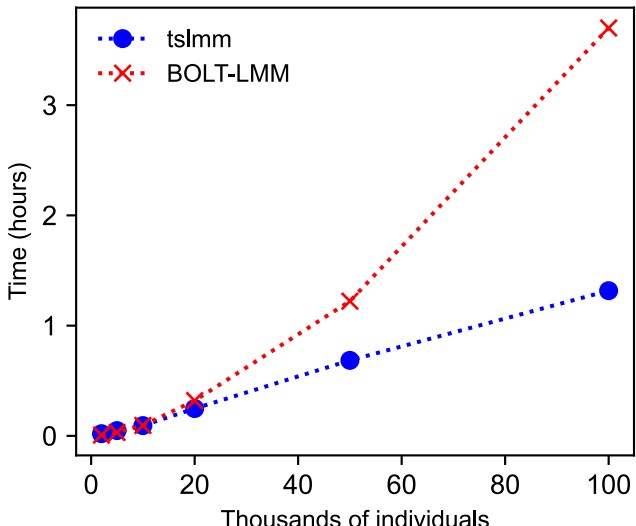

**Fig. 4.** Measured runtimes (on a single thread of Xeon Gold 6,140 CPU) of `tslmm` and `BOLT-LMM` for variance component estimation on 30 simulations of 10 Mbp genome with a recombination rate of $10^{-8}$: five independent simulations at each of six different sample sizes of diploid individuals, summarized with a mean (symbols) and connected with lines.

`BOLT-LMM` or other similar GWAS tools that require variance component estimates.

Although not measured explicitly, the memory demand scales with $\mathcal{O}(KT)$, where $K$ is the number of nodes in the tree sequence and $T$ is the number of test vectors used in randomized trace estimation. This is because a cache array of length $K$ is assigned to each test vector being multiplied to the GRM. As the number of edges in the tree sequence $E$ far exceeds the number of nodes $K$, the algorithm's overhead is unlikely to be significant compared to the size of the original tree sequence that already holds all $E$ edges.

## Accuracy of variance component estimates

To evaluate accuracy of the method we used simulated tree sequences of length 1 Mbp with all other parameters including the

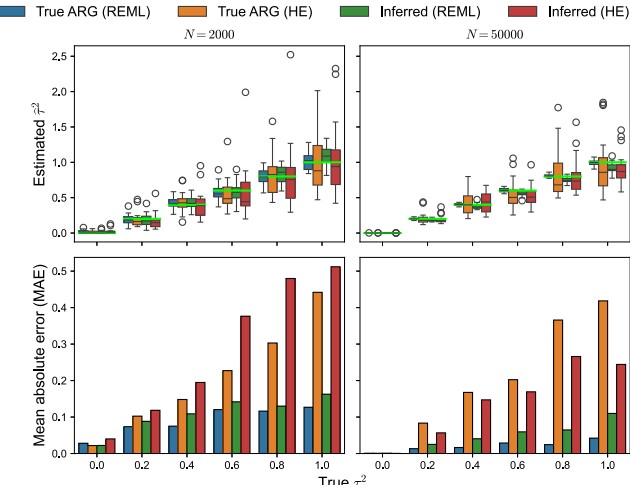

**Fig. 5.** `tslmm` estimates of the genetic variance component $\tau^2$ from true ("True ARG") and inferred ("Inferred") ARGs, using the randomized Haseman–Elston method ("HE") and AI-REML ("REML"), over 200 replicates of 1 Mbp genome with a recombination rate of $10^{-8}$ and $\sigma_\varepsilon^2 = 1$, at each of six different values of $\tau^2 \in \{0.0, 0.2, 0.4, 0.6, 0.8, 1.0\}$. Left column: $N = 2,000$ diploid samples; right column: $N = 50,000$ diploid samples. (Top) Distribution of estimates from the four methods with a standard boxplot: horizontal black line at the median, box extends to the quartiles, whiskers to quartiles $\pm 1.5$ times the interquartile range. The superimposed horizontal line bright green lines denote the true parameter values. (Bottom) The mean absolute error with respect to the true parameter value.

demography as described above. On each tree sequence we simulated phenotypes with $\sigma_\varepsilon^2 = 1$ and a range of values of $\tau^2 \in \{0.0, 0.2, 0.4, 0.6, 0.8, 1.0\}$. Then, we estimated variance components, both using the true and the inferred tree sequences (as described above). We simulated phenotypes, and then inferred variance components, five times on each tree sequence and at each value of $\tau^2$, to quantify the variability of the estimates due to different realizations of simulated phenotypes. Alongside `tslmm`'s main AI-REML routine, we present the performance of the randomized HE method that `tslmm` uses as a starting location for optimization. The relative performance of the two methods is of interest because there is growing interest in using the HE estimator for large-scale quantitative genetics due to its substantially lower computational cost relative to REML-based methods (Wu and Sankararaman 2018; Pazokitoroudi et al. 2020; Loya et al. 2025; Zhu J et al. 2025).

The results are shown in Fig. 5 for $N = 2,000$ and $N = 50,000$ diploids; additional values of $N$ are shown in Figs. A1 and A2. The figure compares estimation on true and inferred tree sequences, as well as the final REML estimates to the estimates produced by the HE method. When the true ARG is available, `tslmm` accurately recovers the variance components, to usually within 10%–20% for the larger sample sizes. Estimates remain nearly as accurate when using an ARG inferred with `tsinfer` and `tsdate` rather than the true ARG. However, as seen in Fig. A1, there is an upwards bias in the inferred $\tau^2$ for the largest sample size ($N = 100,000$ diploids) for larger values of $\tau^2$ (Fig. A1). This bias can also be seen in the mean absolute values (MAE) where `tslmm` on inferred tree sequences has a much larger MAE than on true tree sequences (Fig. A2) despite a comparable variance (Fig. A1). The fact that estimation on inferred ARGs works better with 50,000 than with 100,000 samples (as opposed to using true ARGs, for which the larger sample size has lower MAE), implies

that the bias is due to ARG inference artifacts, which should be correctable given ongoing work on improving ARG inference methods.

Furthermore, the final AI-REML estimates are substantially more accurate than their HE counterparts for both true and inferred trees, with even AI-REML on inferred tree sequences producing better estimates than HE on true tree sequences.

## Relationship to narrow-sense heritability

One major purpose of LMMs is to estimate additive genetic variance and narrow-sense heritability. A standard practice is to obtain the estimates $\tau^2$ and $\sigma_\varepsilon^2$ for the corresponding LMM. Then, one interprets $\tau^2$ as the additive genetic variance and computes the narrow-sense heritability as $h^2 = \tau^2/(\tau^2 + \sigma_\varepsilon^2)$. However, $\tau^2$ in our current model has a different interpretation, and so we describe below this fundamental distinction.

Additive genetic variance of a group of individuals (samples) is defined to be the variance of their (additive) genetic values. If we write $\mathbf{g} = \mathbf{Z}\mathbf{u}$ for these genetic values, then this variance is:

$$V_G = \frac{1}{N}\sum_{i=1}^{N}(\mathbf{g}_i - \overline{\mathbf{g}})^2 = \frac{1}{N}(\mathbf{P}_N\mathbf{g})^T(\mathbf{P}_N\mathbf{g}), \quad (18)$$

where $\mathbf{P}_N = \mathbf{I}_N - \frac{1}{N}\mathbf{1}_N\mathbf{1}_N^T$. This quantity could be treated as an estimate of the population's additive genetic variance, but the accuracy of the estimate will depend on how the sampled individuals in $\mathbf{g}$ represent the population. It's tempting to relate $\tau^2$ and $V_G$, however these quantities differ fundamentally; $\tau^2$ is a *parameter* in the generative model (ARG-LMM), while $V_G$ is an emergent property whose *distribution* depends on $\tau^2$. For instance, the *expected value* of $V_G$ in a diploid Wright-Fisher population of size $N_e$ is $4N_e P\tau^2$ (Lynch and Hill 1986), where $P$ is the number of loci, but genetic drift causes $V_G$ of the population to fluctuate randomly over time. Furthermore, $V_G$ is also impacted by linkage-disequilibrium, which is in turn driven by population processes that impact phenotypes, such as selection and migration (see Lara et al. 2022 and references therein).

How do we estimate $V_G$ from `tslmm`? First note that, still taking the ARG as fixed but the sub-edge effects $\mathbf{u}$ as random, the expected value of the estimator (18) is:

$$
\begin{aligned}
\mathbb{E}[V_G] &= \frac{1}{N}\mathbb{E}\left[(\mathbf{P}_N\mathbf{g})^T(\mathbf{P}_N\mathbf{g})\right] \\
&= \frac{1}{N}\mathbb{E}[\text{tr}(\mathbf{g}^T\mathbf{P}_N^T\mathbf{P}_N\mathbf{g})], \quad \text{because } \mathbf{g}^T\mathbf{P}_N^T\mathbf{P}_N\mathbf{g} \text{ is a scalar} \\
&= \frac{1}{N}\text{tr}(\mathbf{P}_N\mathbb{E}[\mathbf{g}\mathbf{g}^T]\mathbf{P}_N^T), \quad \text{because } \text{tr}(AB) = \text{tr}(BA).
\end{aligned}
$$

Now, because $\mathbb{E}[\mathbf{g}\mathbf{g}^T] = \tau^2\mathbf{B}$, and since $\mathbf{P}_N\mathbf{B}\mathbf{P}_N^T = \widetilde{\mathbf{B}}$, the *centered branch GRM*, we are left with $\mathbb{E}[V_G] = \tau^2\text{tr}(\widetilde{\mathbf{B}})/N$. This suggests that $V_G$ should be close to $\tau^2\text{tr}(\widetilde{\mathbf{B}})/N$. To get an idea of how close (if our model describes reality well), we can compute the variance of this estimator. A similar calculation, provided in Variance of $V_G$, shows that:

$$\text{Var}(V_G) = \frac{\tau^4}{N^2}\sum_{n,m}(\widetilde{\mathbf{B}_{nm}})^2.$$

Since the off-diagonal elements of $\widetilde{\mathbf{B}}$ are $\mathcal{O}(1/N)$, this variance is $\mathcal{O}(1/N)$. Therefore, $\tau^2\text{tr}(\widetilde{\mathbf{B}})/N$ is a consistent estimator of $V_G$, with an error of the order $1/\sqrt{N}$. We raise caution that this calculation

only pertains to panmictic populations. The error can be substantial in practice: for instance, $\mathrm{Var}(V_G)$ is larger in the presence of greater linkage, because the random value of a particular $\mathbf{u}$ will have a larger effect on the realized $V_G$. For more discussion on the estimation of and changes in genetic variance, see Lara et al. (2022).

## Genetic prediction: BLUPs

Once `tslmm` has estimated variance components from data, genetic prediction is relatively straightforward: the typical approach is to use the "best linear unbiased predictor" (BLUP) of the LMM. Suppose we have an ARG that describes relationships between a number of individuals, $N_o$ of which are phenotyped, and $N - N_o$ of which are not. As before, we are working with the model (4), and $\mathbf{y}$ is the vector of all observed phenotype values, but now we wish to find the conditional distribution of the genetic values of a set of individuals whose phenotypic values may or may not have been observed.

Since the model is multivariate Gaussian, this reduces to linear algebra. It is easiest to first explain the general principles that we use, then translate the results into the specific model at hand. Suppose that $\mathbf{x}$ and $\mathbf{z}$ are jointly Gaussian, so that the vector $(\mathbf{x}, \mathbf{z})$ is multivariate Gaussian with mean $\boldsymbol{\mu}$ and covariance $\boldsymbol{\Sigma}$. Write $\boldsymbol{\mu_x}$ and $\boldsymbol{\mu_z}$ for the sub-vectors of $\boldsymbol{\mu}$ giving the means of $\mathbf{x}$ and $\mathbf{z}$, and write $\boldsymbol{\Sigma_{xx}}$, $\boldsymbol{\Sigma_{zz}}$, and $\boldsymbol{\Sigma_{xz}}$ for the respective submatrices of $\boldsymbol{\Sigma}$. Then, the conditional expectation of $\mathbf{x}$ given $\mathbf{z}$ is:

$$\mathbb{E}[\mathbf{x} \mid \mathbf{z}] = \boldsymbol{\mu_x} + \boldsymbol{\Sigma_{xz}}\boldsymbol{\Sigma_{zz}^{-1}}(\mathbf{z} - \boldsymbol{\mu_z}), \tag{19}$$

and the conditional variance of $\mathbf{x}$ given $\mathbf{z}$ is:

$$\mathrm{Var}(\mathbf{x} \mid \mathbf{z}) = \boldsymbol{\Sigma_{xx}} - \boldsymbol{\Sigma_{xz}}\boldsymbol{\Sigma_{zz}^{-1}}\boldsymbol{\Sigma_{zx}}. \tag{20}$$

Note that $\mathbb{E}[\mathbf{x} \mid \mathbf{z}]$ is a linear function of $\mathbf{z}$, and it is a general fact that the conditional mean is an unbiased estimator that minimizes mean squared error, and so $\mathbb{E}[\mathbf{x} \mid \mathbf{z}]$ is, in some sense, the "best linear unbiased predictor" of $\mathbf{x}$ given $\mathbf{z}$.

Now suppose we'd like to estimate the genetic values of a set of unphenotyped individuals. We denote the individuals whose phenotypes we have observed with a subscript $o$, so that $\mathbf{y}_o$ are the observed phenotypes, and $n$ for the non-phenotyped individuals, so that $\mathbf{g}_n$ are their genetic values. (In what follows the two sets of individuals can in fact overlap, but it will be easier to discuss using this convention.) To use the general theory, set $\mathbf{x} = \mathbf{g}_n$, and $\mathbf{z} = \mathbf{y}_o$. Then, the following quantities in equation (19) are:

$$\begin{aligned} \boldsymbol{\Sigma_{xz}} &= \mathrm{Cov}(\mathbf{g}_n, \mathbf{y}_o) = \mathrm{Cov}(\mathbf{g}_n, \mathbf{g}_o) = \tau^2 \mathbf{B}_{n,o} \\ \boldsymbol{\Sigma_{zz}} &= \mathrm{Cov}(\mathbf{y}_o, \mathbf{y}_o) = \tau^2 \mathbf{B}_{o,o} + \sigma_\varepsilon^2 \mathbf{I}_{o,o} \\ \boldsymbol{\mu_z} &= \mathbf{X}_o \mathbf{b} \quad \text{and} \quad \boldsymbol{\mu_x} = \mathbf{0}. \end{aligned} \tag{21}$$

In practice, we replace $\tau^2$, $\sigma_\varepsilon^2$, and $\mathbf{b}$ with their estimates $\hat{\tau}^2$, $\hat{\sigma}_\varepsilon^2$, and $\hat{\mathbf{b}}$ and hence obtain empirical BLUPs (Robinson 1991; Sorensen and Gianola 2002), which are:

$$\widehat{\mathbf{g}}_n = \hat{\tau}^2 \mathbf{B}_{n,o} \big(\hat{\tau}^2 \mathbf{B}_{o,o} + \hat{\sigma}_\varepsilon^2 \mathbf{I}_{o,o}\big)^{-1} \big(\mathbf{y}_o - \mathbf{X}_o \hat{\mathbf{b}}\big). \tag{22}$$

The conditional variances can be calculated using equation (20). The subscripted matrices are the submatrices of the original matrix corresponding to the subscripts. Note that these calculations require access to multiplication with submatrices of $\mathbf{B}$, which can be done as described in Submatrix computation.

In general, we may set $\mathbf{x}$ to any linear combination of sub-edge effects in the tree sequence, as long as we have access to $\boldsymbol{\Sigma_{xz}}$, namely the covariance matrix between the unobserved variable and the observed phenotypic values $\mathbf{y}_o$. The latter provide information about other model variables or their combinations. Hence, $\boldsymbol{\Sigma_{zz}^{-1}}(\mathbf{z} - \boldsymbol{\mu_z})$ is always set to $(\tau^2 \mathbf{B}_{o,o} + \sigma_\varepsilon^2 \mathbf{I}_{o,o})^{-1}(\mathbf{y}_o - \mathbf{X}_o \mathbf{b})$. This is the most compute intensive part of equation (19) but we only need to compute it once for all BLUP predictions. Since $\mathbf{y}_o - \mathbf{X}_o \mathbf{b}$ is a vector, we can obtain $(\tau^2 \mathbf{B}_{o,o} + \sigma_\varepsilon^2 \mathbf{I}_{o,o})^{-1}(\mathbf{y}_o - \mathbf{X}_o \mathbf{b})$ through conjugate gradient as mentioned above. This has many potential applications, such as estimating mutation and edge effects, node values, and individual genetic values, but also their combinations along the genome for genome-wide association studies, over time for studying phenotype evolution, or along the genome and over time to study the origin of the associations. We leave the in-depth investigation of this feature to future work.

## Accuracy of BLUPs

We now assess how well `tslmm`'s BLUPs match the true genetic values, using the same simulated phenotypes and ARGs on 1 Mbp genomes described in Simulations for evaluation of methods. The benchmarks from true tree sequences offer the theoretical performance of `tslmm` provided the ARG is correct, while those from inferred tree sequences indicate the degree to which errors in the ARG introduced by inference affect the quality of the BLUPs. To do this, we first randomly chose half of the individuals as "phenotyped" and set aside the others as "unphenotyped" (but we have true genetic values and phenotypic values for all of them). Using the ARG (either true or inferred, depending on the test), we estimated variance components using only the "phenotyped" individuals, then using these estimates and equation (22), computed BLUPs for the "unphenotyped" individuals.

Figure 6 shows the resulting Pearson correlations between predicted genetic values and the phenotypic values, along with correlations between the true genetic values and the phenotypic values for comparison. Since $\tau^2$ controls the proportion of the heritable component, correlations are larger at larger values of $\tau^2$, simply because more heritable phenotypes are more predictable (recall that the phenotype is equal to genetic value plus environmental component). Thus, even perfect prediction of genetic values (the dotted line in Fig. 6) results in correlations below 1. `tslmm`'s predictions perform only slightly worse than this optimal behavior, but approach perfect prediction at the largest sample sizes ($N = 100,000$). Furthermore, we see only a slight decrease in the phenotype prediction accuracy when using an inferred ARG compared to the true ARG.

## Prediction accuracy with more realistic simulations

The results above are promising, but the simulations we used represent an ideal situation. First, the phenotypes were simulated under the ARG-LMM model (rather than discrete effects associated with mutations). Second, the Wright–Fisher model of demography generates a GRM with very little fine-scale structure, while real data often exhibits fine-scale spatial structure that could in principle affect the accuracy of `tslmm`'s randomized linear algebra methods. We take this opportunity to test the accuracy of phenotype prediction with `tslmm` on a more realistic simulation.

To do this, we simulated a population forward in time using SLiM version 4 (Haller and Messer 2023), under an individual-based model with spatially-restricted mating and dispersal, following Chevy et al. (2025). The simulation takes place in a

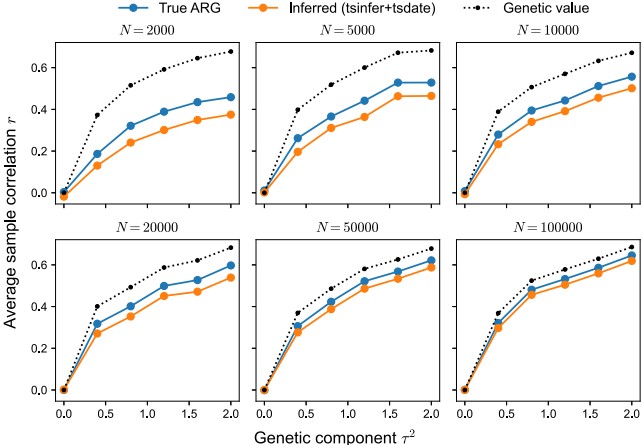

**Fig. 6.** Average Pearson correlation between phenotypic values and `tslmm`'s BLUP of genetic values from true (blue) and inferred (orange) ARGs, using the same simulations as in Fig. 5. The black dotted line is the correlation between the phenotypic value and true genetic value, thereby providing the upper bound of phenotype prediction accuracy. For each simulation, the model was fit using phenotypes from half of the individuals in the ARG, and prediction accuracy is measured using BLUPs from the remaining "held-out" individuals.

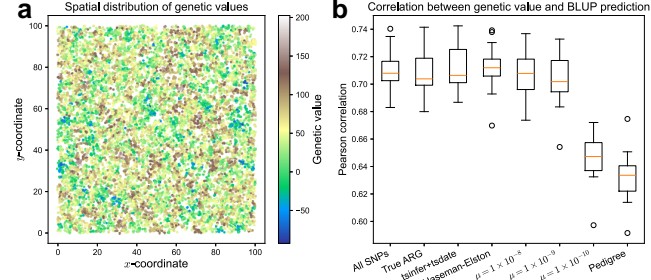

**Fig. 7.** Accuracy of genotype, ARG, and pedigree BLUPs for non-phenotyped individuals in a spatially structured population. a) The spatial distribution of individuals' genetic values, exhibiting spatial autocorrelation. b) The Pearson correlation between BLUPs and true genetic values over 20 replicate simulations, summarized with a boxplot (interquartile range) and median (orange line). The *x*-axis labels refer to the type of GRM: "All SNPs": the standard SNP genotype GRM; "True ARG": `tslmm` using the true ARG; "tsinfer + tsdate": `tslmm` using the ARG inferred by `tsinfer` and `tsdate`; "$\mu$ = value ": the Monte-Carlo SNP genotype GRM (see text) computed from simulated mutations at rate $\mu$ by `arg-needle-lib`; "Pedigree": the pedigree GRM. All methods except `tsinfer +tsdate` used the true ARG, and all methods except `tslmm` obtained estimates by maximizing REML using `nloptr`.

continuous square spatial domain, where the population size is stochastic but kept roughly constant by local competition between nearby individuals. Average dispersal distance and strength of competition were chosen to result in a strong signal of isolation by distance, so that nearby individuals are typically close relatives. This population was simulated for 150 time units (where a time unit corresponds to the expected lifetime of an individual) and the resulting pedigree was recorded. Subsequently, we took all individuals alive at the end of the simulation and simulated a tree sequence backward in time within the pedigree (producing an un-coalesced ARG) using `msprime` with the `fixed_pedigree` model (Anderson-Trocmé et al. 2023). This tree sequence was for a genome of length of $10^8$ bp and a recombination rate of $10^{-8}$. Finally, we "recapitated" the tree sequence by completing the within-pedigree tree sequence backwards in time with the standard coalescent model in `msprime`, using the average population size from the spatial simulation. We repeated both within-pedigree and recapitation coalescent simulations to acquire multiple realizations of the tree sequence from the pedigree. Finally, for each simulated tree sequence, we simulated neutral mutations at a rate of $10^{-8}$, again using `msprime`. This resulted in a total of 13,307 spatially-indexed individuals.

We then simulated the phenotypes using `tstrait` (Tagami et al. 2024) that assigns effects to mutations, instead of **Algorithm T** that assigns effects to sub-edges. One percent of the mutations were set to be causal and their effects were drawn from a standard normal distribution. As before, the resulting genetic value was divided by the square root of the total area up to the roots, so that $\tau^2$ and $\sigma_\epsilon^2$ would be of comparable size. Finally, we returned phenotypic values by adding random environmental deviations from the standard normal distribution to the genetic values. This resulted in a substantial spatial autocorrelation of genetic values, shown in Fig. 7a.

Finally, we computed BLUPs: first, we estimated variance components using phenotypes of 6,654 individuals, and then we obtained BLUPs of genetic values for a separate set of 6,653 "unphenotyped" individuals. Figure 7b shows the accuracy

measured as the correlation between true phenotypes and predicted genetic values; boxplots show the distribution of correlations across replicate simulations. (As in Fig. 6, perfect prediction is not possible because the phenotypes also include the "environmental" deviation.) In this situation, prediction using both the true ARG and an inferred ARG (the second and third boxplots) are very good—as good as the best of Fig. 6, likely because the spatial structuring leads to a more even distribution of close relatives. In this situation, HE method (third boxplot) does roughly as well as AI-REML.

We set up this simulation and chose parameter values in such a way that there was significant local spatial autocorrelation in trait values, and hence local clustering in shared ancestry. This is important because GRMs for long genomes generated under randomly mating demographies tend to be dominated by a few eigenvectors, which leads to faster convergence of the randomized linear algebra methods we use. Strong spatial autocorrelation produces a GRM with a more slowly decaying spectrum, and so provides a stronger test for our computational methods.

## Comparison to genotype, Monte-Carlo GRMs, and pedigree

Finally, we used the same spatial simulations in the previous section to compare `tslmm` with other GRMs for phenotype prediction: the (usual) SNP-derived GRM, the Monte-Carlo estimate of the branch GRM generated by `arg-needle-lib`, and the pedigree-derived GRM. Each uses the same LMM, differing only in which GRM is used. Except `tslmm`, all GRMs were explicitly formed and the variance components were estimated using the eigenvector rotation method for REML (Meyer 1985; Lippert et al. 2011). After multiplying the GRM's eigenvector to the trait and covariates, REML reduces to an independent multivariate Gaussian likelihood optimization. The optimization was carried out by a custom script based on `nloptr`.

The SNP-derived GRM (hereafter, SNP-GRM) is the most common choice of GRM in recent literature (Speed and Balding 2015). The key difference is that while the branch GRM averages over the mutations, the SNP-GRM is constructed using the set of realized variants. Thus, it is of both practical and theoretical

interest to compare the resulting BLUPs. We used the following formula to compute the SNP-GRM between individuals $i$ and $j$:

$$\text{SNP} - \text{GRM}_{ij} = \frac{1}{M} \sum_{k=1}^{M} \frac{(x_{ik} - 2p_k)(x_{jk} - 2p_k)}{2p_k(1 - p_k)},$$

where $x_{ik}$ is the number of copies of the derived variant $k$ in individual $i$, $p_k$ is the frequency of derived variant $k$ across all sampled individuals, and $M$ is the number of realized variants. For application (in Fig. 7) we computed the SNP-GRM after removing variants with minor allele frequency below 0.005.

We also compared our method to the Monte-Carlo estimate of the GRM produced by arg-needle-lib (Zhang et al. 2023). As above, this works by computing the usual genotype GRM after randomly generating mutations on the ARG under the infinite-sites mutation model, whose expectation is the branch GRM. Zhang et al. (2023) showed that this performs well if the mutation rate is high enough, but it is of interest to see how high mutation rate must be, as Zhu J et al. (2025) finds that power and bias are affected by the mutation rate. To do this, we used arg-needle-lib on true tree sequences generated by the simulation, setting $\alpha = 0$ and a range of Monte-Carlo mutation rates.

Comparison to the pedigree is also natural, because plant and animal breeding have long predicted genetic values using a pedigree GRM that computes the expected amount of shared genetic material between relatives (Wright 1922; Cruden 1949; Emik and Terrill 1949). This is done by computing BLUPs in the same way we have described above (Henderson 1984; Lynch and Walsh 1998; Mrode and Pocrnic 2023). These BLUPs are conditional on the observed family structure encoded in the pedigree, while leaving ancestry beyond the pedigree founders unknown, and are hence averaged over. With ARG-LMM we compute BLUPs conditionally on an observed (or inferred) realization of recombinations, mutations and Mendelian segregations since the most recent common ancestors of all sampled individuals, as this is what is recorded in the ARG. We computed the pedigree GRM with the getASubset function from the pedigreeTools package (Vazquez et al. 2018) using Colleau (2002)'s algorithm.

For SNP genotype, Monte-Carlo and pedigree GRMs, we computed BLUPs using nloptr to optimize the REML objective. Accuracies of all methods are shown in Fig. 7b. The SNP-GRM results are comparable to tslmm and Monte-Carlo methods. Results for Monte-Carlo estimated GRMs (labeled by $\mu$, the mutation rate used) show that as previously reported, accuracy is roughly equivalent as for our exact method with higher mutation rate ($\mu = 10^{-8}$), but decreases at lower mutation rates. The final boxplot, labeled "Pedigree," shows that prediction using the pedigree performs substantially less well, as expected.

## Discussion

In this paper, we have presented a collection of methods and open source software that use the ARG to efficiently fit LMMs to population phenotype data, for variance component estimation and genetic prediction. The methods are implemented in the tslmm package and make use of an implicit matrix-vector multiplication algorithm as well as modern randomized linear algebra algorithms to allow computation on large (biobank-scale) datasets. The methods are highly efficient, an efficiency that comes from using the ARG—specifically, the succinct tree sequence encoding—as a sparse matrix representation of the underlying relatedness structure of a population. We have also discussed the

relationship of this ARG-LMM to pedigree and genotype LMMs, and derived a generative model from the underlying mutational model.

Using the ARG allowed us to work with an explicit generative model. As recently reviewed by Lehmann et al. (2026), many of the methods for estimating phenotype variance components and genetic values use essentially the same LMM, differing mostly in choice of GRM. Instead of focusing on the GRM, we focus on the underlying model for genetic values. Sparseness of the resulting model derives from the generative biological processes themselves, and this sparse structure of the model can be leveraged for efficient computations with the LMM (Rue and Held 2005). Although such a structure has been found for pedigree data (Henderson 1976) and has been leveraged for efficient pedigree algorithms (e.g. Meuwissen and Luo 1992; Colleau 2002), finding it for observed genotype matrices has so far been elusive (but see Misztal 2015). In Lehmann et al. (2026) and this paper, we turned to ARG for this structure, because ARGs encode the generative process with past branching/coalescence, recombination, and mutation events.

In addition to computational efficiency, the generative model lets us examine assumptions and points the way towards generalizations. We carefully described the model, including a first-principles derivation from a model of complex traits (Model formulation and assumptions). As one possible extension, one could incorporate information about the locations of causal mutations along the genome, simply by rescaling genomic coordinates so that the length of a region is proportional to the expected proportion of mutational variance the region is responsible for. Similarly, the model assumes that mutations fall uniformly in time, even though this is known not to be the case for mutations under selection and causal mutations are likely under some form of selection. Thus, a readjustment of time—similar to the genotype GRM weighting by functions of frequency—may give better estimates. However, carefully exploring the many strategies is beyond the scope of the current paper.

*Heritability and mutational variance.* The generative model we use has advantages in interpretation over the models used by pedigree and genotype GRMs. With a pedigree, we model variation of genetic values between individuals as a function of their parent average (between family) terms and Mendelian sampling (within family) terms. The variance of these terms in the pedigree model and its GRM is parameterized by the additive genetic variance of the pedigree founders (Henderson 1984; Kennedy et al. 1988; Quaas 1988; Lynch and Walsh 1998; Mrode and Pocrnic 2023). With genotypes, we model variation of genetic values between individuals as a function of their observed genotypes and unobserved allele substitution effects, hence the genotype model and its GRM are parameterized by the variance of allele substitution effects (Whittaker et al. 1997; Meuwissen et al. 2001). There are several variations of the genotype model with a corresponding GRM, which try to connect the variance of allele substitution effects with additive genetic variance (VanRaden 2008; Yang et al. 2010; Speed et al. 2012). However, this connection requires additional assumptions (Gianola et al. 2009; de los Campos et al. 2015; Speed and Balding 2015). Critically, additive genetic variance is by definition a function of both genotypes and allele substitution effects (Falconer 1995; Lynch and Walsh 1998) and there are methods that follow this definition (Hou et al. 2019; Schreck et al. 2019; Lara et al. 2022; Feldmann et al. 2023). Our analysis matches this definition; we show that additive genetic variance depends on the sample at hand, because genotype distributions

vary between samples. With the ARG, we model variation of genetic values between individuals as a function of edge effects, hence the ARG model and its GRM are parameterized by the variance of edge effects ($\tau^2$). We can estimate this generative parameter (with units of variance generated per generation) thanks to the fact that the ARG lets us express relatedness in units of generations—another example of estimated ARGs providing a time axis for genomic data.

We also discussed estimation of the additive genetic variance for a given sample. In some models, mutational variance and additive genetic variance have a closer relationship. For instance, it is common to define the relatedness matrix as $\mathbf{X}\mathbf{X}^T$, where $\mathbf{X}_{ij} = (\mathbf{x}_{ij} - p_j)/(p_j(1-p_j))^\alpha$ (or $\mathbf{X}_{ij} = (\mathbf{x}_{ij} - 2p_j)/(2p_j(1-p_j))^\alpha$ in diploids), where $\mathbf{x}_{ij}$ is the number of derived alleles at locus $j$ carried by individual $i$, $p_j$ is the derived allele frequency in the sample at locus $j$, and $\alpha = 1/2$ (VanRaden 2008; Yang et al. 2010). Then, for haploids or if Hardy-Weinberg equilibrium holds, $\mathrm{tr}(\mathbf{X}\mathbf{X}^T)$ is equal to $M$, the number of SNPs, provided the linkage-disequilibrium across loci is sufficiently weak. In this case, the two quantities ($\tau^2$ and $\mathbb{E}[V_G]$) are essentially the same, differing only by a factor of $M$. However, this is not the case in our model, nor would it be if $\alpha \neq 1/2$ is a good description of reality.

*Unobserved mutations and GRM's predictive performance.*
Although the branch GRM naturally arises from a mutational model, the choice to use the branch GRM can be counterintuitive. This is because the branch GRM treats the ARG as fixed while treating the number and locations of the causal mutations on the ARG as random, even though mutations are more directly observed (but not necessarily the causal mutations). To clarify this relationship, we give careful consideration to these assumptions, including a more mathematical treatment in the Appendix. However, there are many variants that are not represented in typical genetic data: for instance, many structural variants, rare alleles without sufficient genotyping, or, for studies using genotyping arrays, simply variants not included on the array. The ARG has the potential to represent these ungenotyped variants (if the branches they fall on appear in the inferred ARG), and so the ARG-LMM model could work well for complex traits whose causal mutations are unknown and perhaps ungenotyped. Indeed, Zhu J et al. (2025) demonstrated that in practice the ARG could be used to identify additional signal not directly represented by genotypes.

Even if we do perfectly observe the number and locations of mutations, their effect sizes are still undetermined, and so the random effects models one uses in the two cases are very similar. If we have strong prior knowledge of which alleles are causal, and these are all genotyped, a corresponding genotype GRM just with these sites is expected to work better (although computing with it using an ARG could be very efficient). In practice, however, our information about allelic effects only allows a weak prior. A mathematically appealing feature of treating mutations as random is that genetic fixed effects all collapse to an intercept, alleviating the burden of modeling fixed effects. Which approach works better in practice (or, perhaps a mix of the two) remains an empirical question.

*Limitations and robustness.* More generally, there are many directions to explore how this might be best applied to real data. One such factor is the ARG inference method, including data pre-processing steps and choice of software. It will be important for future work to explore best practices for ARG inference, including data collection, cleaning, phasing, imputation, ancestral state polarization, parallelization across segments of the genome, algorithmic parameter tuning, mutation rate calibration for dating, and quality diagnostics. Which methods perform best may depend strongly on the data, in particular on sample size, genetic diversity, error rates, and type of genotyping, and so studies at just one scale or with just one data type run a risk of over-generalization.

Of course, errors in ARG estimation are inevitable, and so the ideal ARG-based estimator would be unbiased, so that effects of these errors "average out" in the downstream estimates. Furthermore, the Gaussian model for edge effects may be a better choice for some traits than others.

Some aspects of the method suggest the results will be robust. For instance, our derivation of the AI-REML algorithm is based on a Gaussian model, even though mutations appear on branches in a discrete fashion. However, Harville (1977) and Westfall (1987) showed that REML coincides with minimum norm quadratic unbiased estimation (MINQUE), which is a moment-based estimator. Hence, normality is not a requirement for consistent point estimation, provided the covariance matrix $\mathbf{V}$ is correct. Indeed, tslmm returned consistent variance component estimates and BLUPs in the spatial benchmark, where the phenotype was generated from discrete mutations by tstrait (Fig. A3).

Another way things can go wrong in practice is in convergence of estimates, but results here are again encouraging. Above, we saw that there is a fortuitous cancelation in the average information matrix that may contribute substantially to the efficiency of stochastic optimization. The Hessian (equation 12) and the Fisher information (equation 13) each have a trace term, but these cancel in average information (equation 14). This is not only computationally convenient, but it also means that we do not rely on stochastic estimates of the curvature, which can be problematic in practice for second-order optimization methods.

*Relationship to Zhu J et al. (2025).* Some of the aims of this paper overlap with those of the concurrent work in Zhu J et al. (2025), which describes arg-lmm, a linear mixed model software built on top of the GRM-vector multiplication algorithm of arg-needle-lib. Both papers use the ARG for fast linear algebra to fit variance components of complex traits. Looking at the big picture, Zhu J et al. (2025) focuses on identifying regions associated to traits like GWAS, while this work studies genetic prediction. In detail, Zhu J et al. (2025) goes much further than we do in application, including analysis of many real traits in an ARG derived from real data, while we study the generative model (ARG-LMM), focus on algorithmic and theoretical improvements, and test these in two simulations with variance component estimation and genetic prediction.

The ARG-powered GRM-vector multiplication algorithms lie at the heart of both studies. The key difference is that we perform exact computations with the branch GRM, while Zhu J et al. (2025) use a Monte Carlo approximation that randomly places mutations on branches proportional to their area. As shown in Zhu J et al. (2025) (as well as in Fan et al. 2022; Zhang et al. 2023), this provides highly accurate estimates as long as the number of mutations is large enough. We have shown that the Monte Carlo estimation is unnecessary, as exact values can be computed in a fraction of the time.

In terms of variance component estimation, both arg-lmm and tslmm implement the Haseman-Elston (HE) method, but tslmm only uses HE results as a starting location for optimizing the REML. We estimate variance components for the whole genome to conduct genetic prediction, while Zhu J et al. (2025) perform it

locally for GWAS. Finally, to compute the traces that play a central role in variance component estimation, Zhu J et al. (2025) use the Hutchinson estimator that has an error with variance $1/n$ (where $n$ is the number of random test vectors used; Hutchinson 1990); we use a newer estimator that has variance $1/n^2$ (XTrace—Epperly et al. 2024), and so in practice needs many fewer test vectors.

*Future directions.* Rebollo et al. (2025) and Fortuna et al. (2025) study the estimation of haplotype and mutation effects on local trees. The machinery presented in this paper generalizes their tree-by-tree approach, allowing to compute any linear combination of sub-edge effects at scale. Such application includes time-resolved association testing (Link et al. 2023; Zhang et al. 2023; Zhu J et al. 2025) and estimating ancestral genetic values (Edge and Coop 2018; Peng et al. 2025). Frequency-dependent weighting of branch GRM, available in other software (Fan et al. 2022; Zhang et al. 2023; Zhu J et al. 2025), is yet to be implemented in `tskit` (but would not require a fundamental modification of the GRM-vector multiplication algorithm).

*In closing.* The best way to model and predict complex traits doubtless depends on many factors, including the type of data available and the evolutionary forces on the trait. The results we have shown here demonstrate two of the primary advantages to working with ARGs: (1) they provide a "time axis" (which here led to a more interpretable variance component parameter $\tau^2$, in units of mutational variance per generation), and (2) they are extremely compact, so allow easy computation with large datasets.

## Data availability

`tslmm` is a Python package available at https://github.com/hanbin973/tslmm. The results and the figures of the paper can be reproduced using the scripts at https://github.com/hanbin973/tslmm_paper.

Supplemental material available at GENETICS online.

## Acknowledgments

Many thanks to Roshni Patel for explaining to us some aspects of heritability, to Doc Edge for discussion about an early idea of the paper, and to Yan Wong and Jonathan Terhorst for helpful comments and suggestions on a draft. We also thank the participants of the Probabilistic Modeling in Genomics 2025 conference for valuable questions and discussions. This research was supported in part through computational resources and services provided by Advanced Research Computing at the University of Michigan, Ann Arbor.

## Funding

HL acknowledges the support from the Department of Statistics, University of Michigan, Ann Arbor through the Departmental Fellowship. NP, JK, and PR acknowledge support from the NIH NHGRI (research grant HG012473). GG acknowledges support from the BBSRC ISP grant to The Roslin Institute (BBS/E/D/30002275, BBS/E/RL/230001A, and BBS/E/RL/230001C), BBSRC research grant BB/T014067/1, and the NRC research grant 346741.

## Conflicts of interest

The author(s) declare no conflicts of interest.

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

# Appendix

## Appendix A: Model formulation and assumptions

### Expressing a complex trait on an ARG

We show the correspondence between site and branch (edge) based trait expressions. We define an ARG $\mathcal{T} = (\mathcal{N}, \mathcal{E})$ as a collection of nodes $\mathcal{N}$ and edges $\mathcal{E}$. Each node $n$ has a (birth) time ago $t_n$, and each edge $e$ indicates inheritance of a genomic interval $[\ell_e, r_e)$ by a child $c(e)$ from a parent $p(e)$ (Wong et al. 2024). We call these "parent" and "child" in graph terms, so the inheritance might be over more than one generation. To describe a valid ARG, the spans of all edges with the same child must be disjoint, and the time of the parent must be larger than the time of the child for each edge. We will also call the *span* of an edge $s_e = r_e - \ell_e$ (in number of base-pairs), and the *length* of an edge $l_e = t_{p(e)} - t_{c(e)}$ (in time units, such as generations). All expectations and variances are conditioned on $\mathcal{T}$, which accounts for the variation in observed genome data due to recombination and drift, and hence also the implicit population structure and pedigree of individuals in $\mathcal{T}$. Only the mutational process is left random in this setting. This setting was explored in Ralph (2019) and expanded in Ralph et al. (2020). In those papers, the utility of treating mutations as random on a fixed ARG was to understand how well a given SNP genotype (site) statistic could be expected to estimate the corresponding summary of tree shapes.

For conceptual clarity, we assume in the following that the edges are in fact *sub-edges*: each has a unique sub-topology along its span. (Although this is not true in the case of tree sequences in general, we can enforce this by appropriate splitting of edges as in Nowbandegani et al. 2023.) Nevertheless, this splitting is merely conceptual and never takes place in the algorithms we'll demonstrate shortly, as it significantly degrades the storage efficiency of the tree sequence data structure.

We consider complex traits that follow a linear model. Write $\mathbf{G} \in \mathbb{R}^{N \times P}$ for the $N \times P$ genotype matrix for $N$ genomes and $P$ sites in the genome, whose $(i, j)$th entry is the number of derived alleles carried by the $i$th genome at the $j$th site. Then the trait model is:

$$\mathbf{y} = \mathbf{G}\boldsymbol{\beta} + \boldsymbol{\varepsilon} = \sum_{p=1}^{P} \mathbf{G}_p \boldsymbol{\beta}_p + \boldsymbol{\varepsilon}, \tag{A1}$$

where $\boldsymbol{\beta} \in \mathbb{R}^P$ is a vector defining a effect size for the derived allele(s) at each site. For now, we condition on $\boldsymbol{\beta}$ so that we treat it as a fixed quantity. With infinite-sites mutation model, each

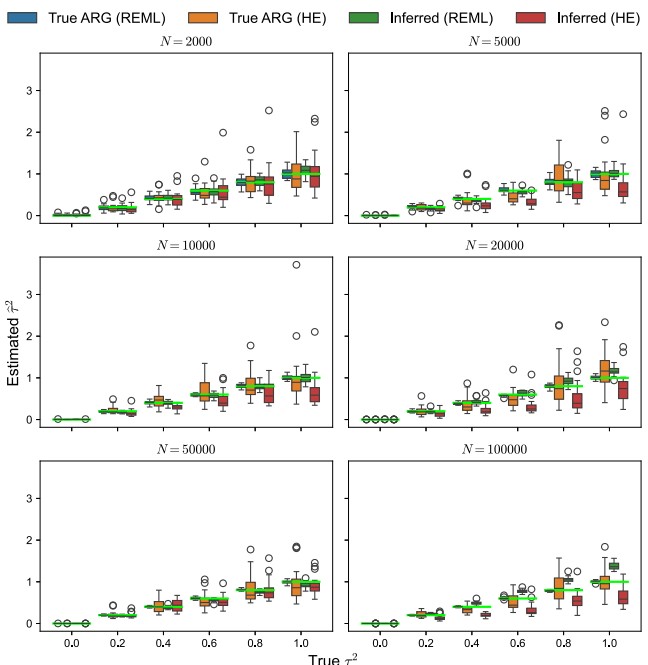

**Fig. A1.** Statistical precision of the HE method and AI-REML. By default, the branch GRM is scaled by 1 over $4 \sum_T s_T t_{\text{root},T}$. The values in the labels "HE (value)" refer to the additional scaling we apply to the GRM. Each boxplot shows the distribution of estimators over 100 replicates.

site has at most two alleles, so this effect is the allele substitution effect (Falconer 1995), also called the site effect, variant effect, or mutation effect in this work. One cannot estimate $\boldsymbol{\beta}$'s entries separately because some variants appear on the same edge (or on the two edges pendant to the root). Columns of $\mathbf{G}$ corresponding to these variants will be identical, leading to perfect collinearity (linkage-disequilibrium), making them statistically indistinguishable. We will group these indiscernible variants according to their edge membership. After conditioning on $\mathcal{T}$, the remaining variation in $\mathbf{G}$ is solely due to the mutational process. We assume that mutations occur on edge $e$ following a Poisson process at a rate proportional to the total number of base-pairs (span) multiplied by the amount of time the edge represents (length), $l_e s_e$. We let the per unit time per base-pair mutation rate $u_{ep}$ vary across edges and positions If we let $\mathbf{1}_{ep}$ be an indicator variable that is 1 if a mutation has arrived on edge $e$ at position $p$ and 0 otherwise, $\mathbb{E}[\mathbf{1}_{ep}] = l_e u_{ep}$.

Recurrent mutations are allowed; however, mutations cannot occur below an existing one. This is a slight relaxation of the infinite-sites model to allow multiple recurrent mutations in unrelated lineages but no back mutation. Define $\mathbf{Z} \in \mathbb{R}^{N \times E}$ so that $\mathbf{Z}_{ie}$ is the number of chromosomes of individual $i$ that is a descendant of edge $e$. $\mathbf{Z}_{ie}$ ranges from 0 to 2 in the case of diploids. We can express $\mathbf{G}$ with $\mathbf{Z}$ by:

$$\mathbf{G}_{ip} = \sum_{e\,:\,p \in e} \mathbf{Z}_{ie} \mathbf{1}_{ep}, \tag{A2}$$

where $p \in e$ means edge $e$ contains the position $p$, and define $\mathbf{G}_p$ as the $p$th column of $\mathbf{G}$. In words, one can determine individual $i$'s genotype at position $p$ by first examining if edge $e$ is ancestral to $i$ ($\mathbf{Z}_{ie}$) and then checking if the edge has a mutation at $p$ ($\mathbf{1}_{ep}$). Finally, we sum up such contributions across all edges ($\sum_{e\,:\,p \in e}$). This also illustrates how `tstrait` computes the trait values:

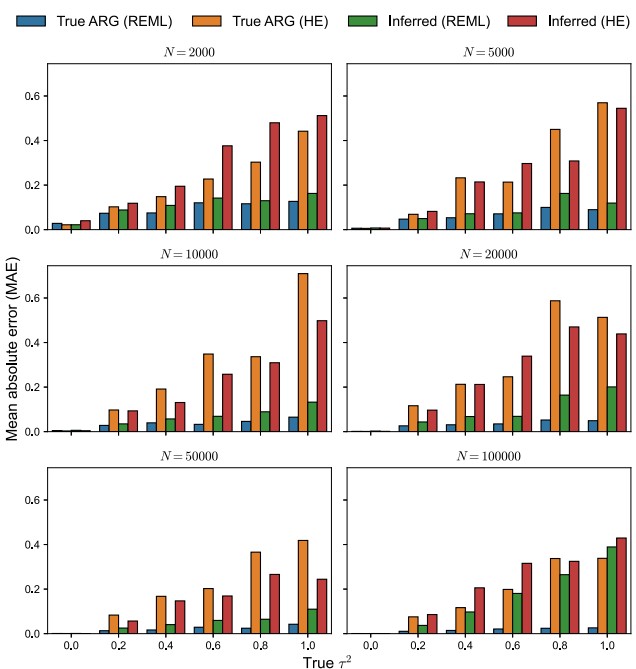

**Fig. A2.** Statistical precision of the HE method and AI-REML. By default, the branch GRM is scaled by 1 over $4\sum_T s_T t_{\text{root},T}$. The values in the labels "HE (value)" refer to the additional scaling we apply to the GRM. The mean absolute error (MAE) of the estimators with respect to the true parameters $\sigma_\varepsilon^2 = 1$ and $\tau^2 = 1$.

for a position $p$ in a local tree, it travels down the edges (or equivalently branches for now) of the tree and checks if there's a mutation along the path from the root to the tips. Here, $\mathbf{Z}_{ie}$ tells if an edge in $\{e : p \in e\}$ is on the path from the root to a tip, which is individual $i$.

Using equation (A2), we can collapse variants corresponding to perfectly correlated columns of $\mathbf{G}$ into the edge on which they reside. This is a natural choice because mutations on the same edge are inherited by the same set of individuals (recall that the edge's subtree is constant by assumption). Substituting (A2) in the linear equation (A1) gives:

$$
\begin{aligned}
\mathbf{y} &= \sum_{p=1}^{P} \mathbf{G}_p \boldsymbol{\beta}_p + \boldsymbol{\varepsilon} \\
&= \sum_{p=1}^{P} \left( \sum_{e : p \in e} \mathbf{Z}_e \boldsymbol{\beta}_p \mathbf{1}_{ep} \right) + \boldsymbol{\varepsilon} \\
&\quad \because \text{substitute(A2)} \\
&= \sum_{e=1}^{E} \sum_{p : p \in e} \mathbf{Z}_e \boldsymbol{\beta}_p \mathbf{1}_{ep} + \boldsymbol{\varepsilon} \\
&\quad \because \text{swap the sums over } e \text{ and } p \\
&= \sum_{e=1}^{E} \mathbf{Z}_e \left( \sum_{p : p \in e} \boldsymbol{\beta}_p \mathbf{1}_{ep} \right) + \boldsymbol{\varepsilon} \\
&\quad \because \mathbf{Z}_e \text{ does not depend on } p \\
&= \sum_{e=1}^{E} \mathbf{Z}_e \mathbf{v}_e + \boldsymbol{\varepsilon} \quad \because \mathbf{v}_e = \sum_{p : p \in e} \boldsymbol{\beta}_p \mathbf{1}_{ep} \\
&= \mathbf{Z}\mathbf{v} + \boldsymbol{\varepsilon}.
\end{aligned}
\tag{A3}
$$

The key step is at the third line which swaps the summation over edges $e$ and sites $p$. Then, we pulled out $\mathbf{Z}_e$ out of the inner sum over $p$ and merged the inner sum into $\mathbf{v}_e$. This derivation presents

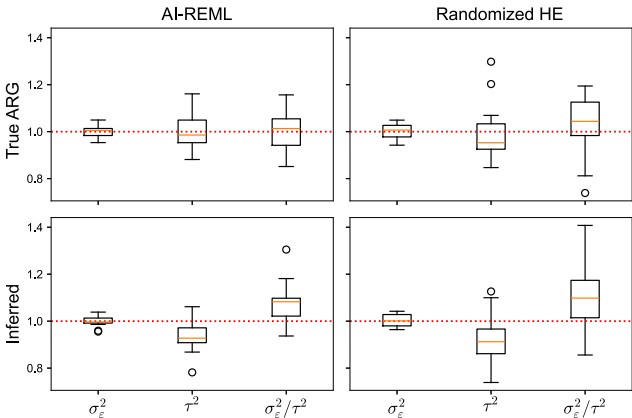

**Fig. A3.** Statistical precision of the HE method and AI-REML in the spatial simulation benchmark. 20 replicate experiments were conducted.

a way to convert a genotype-based expression of a trait $\mathbf{y} = \sum_p \mathbf{G}_p \boldsymbol{\beta}_p + \boldsymbol{\varepsilon}$ into a branch-based expression $\mathbf{y} = \sum_e \mathbf{Z}_e \mathbf{v}_e + \boldsymbol{\varepsilon}$.

## Linear algebra formulation

We now connect this linear model to classical formulations in quantitative genetics and selective breeding literature on the pedigree model (Henderson 1976; Quaas 1988). While we work with a linear model on an ARG, not pedigree, our aim here is to highlight the inherent sparse structure of phenotype generative models and the critical connection with sparse linear algebra and efficient computational algorithms (Rue and Held 2005). We start by noting that sub-edges are also edges—each has a parent and child node and a genomic interval—but with the property (as above) that the subtree below them in the ARG does not change. These can be obtained as follows: iterating along the genome, each time the local tree changes, it changes through addition or removal of an edge. If the parent of the newly added or removed edge is $n$, then we move up through the local tree, and split each edge we encounter. For instance, if $n$ is not a root then there is an edge with child $n$ and parent $p$, and interval $[\ell, r]$ that spans the current position $x$. By "splitting" this edge we mean to replace it with two edges: one with interval $[\ell, x)$ and the other with interval $[x, r)$. The resulting shorter edges will be all sub-edges.

We say that sub-edge $e_1$ *directly inherits* from sub-edge $e_2$ if $p(e_1) = c(e_2)$ and their genomic intervals overlap. In fact, it then follows that the interval of $e_2$ is contained within that of $e_1$: in other words, $[\ell(e_2), r(e_2)) \subseteq [\ell(e_1), r(e_1))$. Now, define the square matrix $\mathbf{A}$ to have rows and columns indexed by sub-edges, with $\mathbf{A}_{ef} = 1$ if sub-edge $e$ directly inherits from $f$, and 0 otherwise. This $\mathbf{A}$ is $\mathbf{P}$ in Quaas (1988). The *sub-edge inheritance matrix* is then $\mathbf{I} + \mathbf{A} + \mathbf{A}^2 + \cdots = (\mathbf{I} - \mathbf{A})^{-1}$: this has $(e, f)$th entry equal to 1 if $e = f$ or if $e$ inherits from $f$ (i.e. if the edge represented by $e$ carries genetic material inherited also along $f$), and 0 otherwise. Note that the seemingly infinite series above is in fact a finite sum because the depth of an ARG is finite. Now, write $\mathbf{M}$ for the *individual-sub-edge incidence matrix*: for an individual $i$ and a sub-edge $e$, let $\mathbf{M}_{ie} = 1$ if $c(e) \in i$ and $\mathbf{M}_{ie} = 0$ otherwise. $c(e) \in i$ means individual $i$ contains node $c(e)$. The matrix $\mathbf{Z}$ from the main text is thus:

$$
\mathbf{Z} = \mathbf{M}(\mathbf{I} - \mathbf{A})^{-1}.
$$

The directed graph whose adjacency matrix is $\mathbf{A}$, the *sub-edge graph*, is a multitree, and has many convenient properties.

For instance, many standard ARG computations that require left-to-right iteration can be implemented as bottom-up or top-down on the sub-edge graph. This property is shared by the genome representation graph of DeHaas et al. (2024). Typically, there will be a multiplicative factor of order $\log N$ more sub-edges than edges, where $N$ is the sample size. However, this difference can be large in practice as the constant depends on demographic history. An advantage to working with the sub-edge graph is that it allows use of more general graph-based tools, as for instance by Nowbandegani et al. (2023). However, we are not aware of a reason that such algorithms are necessarily more efficient.

This formulation is useful conceptually, but we did not find it to be useful computationally, because typical left-to-right algorithms on the ARG naturally work with sub-edges without explicitly constructing them (for instance, those in Ralph et al. 2020 or Zhu J et al. 2025). We are not aware of any argument that algorithms using a structure like the sub-edge graph are inherently more efficient than algorithms that efficiently use the tree sequence.

Sub-edges, and the "splitting" procedure are very similar to the "bricks" and the "bricking" operation used by Nowbandegani et al. (2023). However, "bricks" are defined so that they have a unique set of descendant nodes, rather than a unique subtree. Since subtrees can change but still have the same set of descendants, bricks are thus larger than sub-edges, while retaining the useful properties above. However, the brick graph is no longer a multitree. We chose the more strict definition mainly because it does not require retaining and checking the descendant nodes at each step.

The linear algebra formulation shows the sparse structure of our trait generative model on an ARG, enabling scalable algorithms as shown in Lehmann et al. (2026) and this study. For completeness, we point to Misztal (2015), who followed a similar goal —enabling inexpensive computations with the inverse of genotype GRM by avoiding inversion of a large GRM. Instead of focusing on the trait generative model, Misztal (2015) split the genotyped sample into core and non-core sets of individuals and expressed genetic values of non-core individuals conditional on core individuals. This conditioning also admits a sparse inverse of genotype GRM and is technically the same as conditioning on "inducing point" or "knots" in spatial modeling (e.g. Heaton et al. 2018). While this core conditioning approach is computationally very efficient for populations with limited genetic diversity, such as in selective breeding, it does not scale as well to more diverse settings as the work presented in this study.

## The branch-based phenotype model is a linear mixed model

The last expression we arrived at expresses a phenotype in terms of edges:

$$\mathbf{y} = \sum_e \mathbf{Z}_e \mathbf{v}_e + \boldsymbol{\varepsilon}. \tag{A4}$$

We refer to $\mathbf{v}_e$ as the edge effect and we shall characterize its probabilistic features such as its distribution. We go back to the definition of the edge effect and observe that it is a collection of indicator random variables driven by mutation:

$$\mathbf{v}_e = \sum_{p\,:\,p \in e} \boldsymbol{\beta}_p \mathbf{1}_{ep}. \tag{A5}$$

In turn, $\mathbf{v}_e$ as a whole is a random variable. Since mutations occur at a low rate, $\mathbf{v}_e$ is likely to be zero most of the time,

but conditioning on the occurrence, a mutation hits position $p$ at a probability that is proportional to the position's mutation rate $u_{ep}$. The effect size of this mutation is $\boldsymbol{\beta}_p$. Altogether, $\mathbf{v}_e$ is zero or a number drawn from the pool of effect sizes $\{\boldsymbol{\beta}_p\}_{p \in e}$ with probability proportional to the mutation rate $u_{ep}$. Hence, the distribution is determined by the collection $\{(\boldsymbol{\beta}_p, u_{ep})\}_{p\,:\,p \in e}$.

This formulation closely mirrors the continuum-of-alleles (CoA) model proposed by Kimura and Crow (1964). Similar to an edge that contains many positions, a locus also stretches over a genome segment that is subject to mutation. When a mutation arrives, it occurs at a random position yielding a different allele every time, i.e. the infinite-alleles model. The CoA model describes the distribution of the effect size of these newly emerging alleles.

Our branch based expression in equation (A4) represents a phenotype by a design matrix with effect sizes that are random with respect to mutations. Hence, it can be considered as a linear random effects model. Furthermore, some features of the model can be deduced from the properties of the mutational process and the topology of the ARG. The variance of an edge can be computed as follows:

$$
\begin{aligned}
\mathrm{Var}(\mathbf{v}_e) &= \mathrm{Var}\left( \sum_{p\,:\,p \in e} \boldsymbol{\beta}_p \mathbf{1}_{ep} \right) \\
&= \sum_{p\,:\,p \in e} \boldsymbol{\beta}_p^2 \mathrm{Var}(\mathbf{1}_{ep}) \\
&\quad \because \text{mutations at different } p \text{ are independent} \\
&= \sum_{p\,:\,p \in e} \boldsymbol{\beta}_p^2 l_e u_{ep}(1 - l_e u_{ep}) \\
&\quad \because \mathrm{Var}(\mathbf{1}_{ep}) = \mathbb{E}[\mathbf{1}_{ep}](1 - \mathbb{E}[\mathbf{1}_{ep}]). \\
&\approx \sum_{p\,:\,p \in e} \boldsymbol{\beta}_p^2 l_e u_{ep} \\
&\quad \because l_e u_{ep} \ll 1 \text{i.e., mutation rate is small} \\
&= \overbrace{l_e s_e}^{\text{edge area} := A_e} \cdot \overbrace{\frac{1}{s_e} \sum_{p\,:\,p \in e} \boldsymbol{\beta}_p^2 u_{ep}}^{\substack{\text{size average of squared} \\ \text{effect sizes times} \\ \text{mutation rate: } = \tau_e^2}} \\
&\quad \because l_e \text{ does not depend on } p \\
&= A_e \tau_e^2
\end{aligned}
\tag{A6}
$$

The variance of an edge effect is proportional to its area $A_e$. $\tau_e^2$ is the average squared effect size weighted by mutation rates. In words, regions with higher functional importance (large $\boldsymbol{\beta}_p^2$) and mutation rate (large $u_{ep}$) contribute more to branch/edge effect variation. Assuming that mutation rates are constant and the effect sizes are evenly dispersed across the genome, similar to the infinitesimal model, this recovers the ARG-LMM model we have used in the main portion of this paper.

In the main portion of the paper, the branch effects were assumed to be mean zero and mutually independent. A derivation of these (approximate) properties helps shed light on how and when this is a good assumption. To do this, we decompose $\mathbf{v}_e$ into two parts $\mathbf{u}_e = \mathbf{v}_e - \mathbb{E}[\mathbf{v}_e]$ and $\mathbf{f}_e = \mathbb{E}[\mathbf{v}_e]$. The vector of these quantities are defined accordingly: $\mathbf{u} = \mathbf{v} - \mathbb{E}[\mathbf{v}]$ and $\mathbf{f} = \mathbb{E}[\mathbf{v}]$. Then, we can rewrite equation (A4) as:

$$\mathbf{y} = \mathbf{Z}\mathbf{f} + \mathbf{Z}\mathbf{u} + \boldsymbol{\varepsilon}. \tag{A7}$$

Henceforth, we refer to $\mathbf{u}_e$ instead of $\mathbf{v}_e$ as the edge effect. From the definition, the branch effect $\mathbf{u}_e$ has mean zero. At first glance, enforcing zero mean by subtracting the mean looks like an

unfounded assumption that sweeps the mean contributions $\mathbf{Zf}$ under the rug. Surprisingly, if all individuals exist at the same time, this is not an issue. Under neutral evolutionary scenarios of various kinds, we may assume that the mutation rate is constant for each position (Ewens 2004; Durrett 2008; Wakeley 2008), i.e. $u_{ep} = u_p$ for all $e$ such that $p \in e$ (edge $e$ contains position $p$). In this case, $\mathbf{Zf}$ reduces to a constant:

$$
\begin{aligned}
[\mathbf{Zf}]_i &= \sum_{e=1}^{E} \mathbf{Z}_{ie} \mathbb{E}\left[ \sum_{p:p\in e} \boldsymbol{\beta}_p \mathbf{1}_{ep} \right] \\
&\quad \because \mathbf{f}_e = \mathbb{E}[\mathbf{v}_e] \\
&= \sum_{e=1}^{E} \sum_{p:p\in e} \mathbf{Z}_{ie} \boldsymbol{\beta}_p l_e u_p \\
&\quad \because \mathbb{E}[\mathbf{1}_{ep}] = l_e u_{ep} \quad \text{and} \quad u_{ep} = u_p \\
&= \sum_{p=1}^{P} \sum_{e:p\in e} \mathbf{Z}_{ie} \boldsymbol{\beta}_p l_e u_p \\
&\quad \because \text{swap the sums over } e \text{ and } p \\
&= \sum_{p=1}^{P} \boldsymbol{\beta}_p u_p \left( \sum_{e:p\in e} \mathbf{Z}_{ie} l_e \right) \\
&\quad \because \boldsymbol{\beta}_p \text{ and } u_p \text{ do not depend on } e \\
&= \sum_{p=1}^{P} \boldsymbol{\beta}_p u_p \cdot 2 t_{\text{root},p} \quad \because \text{ploidy} = 2 \\
&= \text{constant with respect to } i.
\end{aligned}
\tag{A8}
$$

Note that $\sum_{e:p\in e} \mathbf{Z}_{ie} l_e$ is the sum of lengths of all edges ancestral to individual $n$ that adds up to the root time at position $p$ multiplied by ploidy. This means that the fixed component of the genetic effect is fully accounted for by a model intercept.

We can further generalize equation (A8) to incorporate time varying mutation rates. In this setting, we stipulate that a mutation rate at a position only depends on time, i.e. the mutation rate is a function of time $u_p(t)$:

$$
\begin{aligned}
[\mathbf{Zf}]_i &= \sum_{e=1}^{E} \mathbf{Z}_{ie} \mathbb{E}\left[ \sum_{p:p\in e} \boldsymbol{\beta}_p \mathbf{1}_{ep} \right] \\
&= \sum_{e=1}^{E} \sum_{p:p\in e} \mathbf{Z}_{ie} \boldsymbol{\beta}_p \mathbb{E}[\mathbf{1}_{ep}] \\
&\quad \because \mathbf{Z}_{ie} \text{ does not depend on } p \\
&= \sum_{e=1}^{E} \sum_{p:p\in e} \mathbf{Z}_{ie} \boldsymbol{\beta}_p \int_{t(e(c))}^{t(e(p))} u_p(t) dt \\
&\quad \because \mathbb{E}[\mathbf{1}_{ep}] = \int_{t(e(c))}^{t(e(p))} u_p(t) dt \\
&= \sum_{p=1}^{P} \sum_{e:p\in e} \mathbf{Z}_{ie} \boldsymbol{\beta}_p \int_{t(e(c))}^{t(e(p))} u_p(t) dt \\
&\quad \because \text{swap the sums over } e \text{ and } p \\
&= \sum_{p=1}^{P} \boldsymbol{\beta}_p \sum_{e:p\in e} \mathbf{Z}_{ie} \int_{t(e(c))}^{t(e(p))} u_p(t) dt \\
&\quad \because \boldsymbol{\beta}_p \text{ does not depend on } e \\
&= \sum_{p=1}^{P} \boldsymbol{\beta}_p \left( 2\int_{0}^{t_{\text{root},p}} u_p(t) dt \right) \quad \because \text{ploidy} = 2 \\
&= \text{constant. w.r.t. } i.
\end{aligned}
\tag{A9}
$$

This confirms that the fixed portion $\mathbf{Zf}$ of equation (A7) is constant in a fairly general setting with time varying mutation rates.

In order to demonstrate independence, we show that $\text{Cov}(\mathbf{u}_e, \mathbf{u}_{e'}) = 0$ for all pairs $e \neq e'$. This implies independence because indicator variables are independent if and only if they are

uncorrelated, similar to Gaussian random variables. We first expand $\text{Cov}(\mathbf{u}_e, \mathbf{u}_{e'})$ and reduce it to a sum of covariances of indicators:

$$
\begin{aligned}
&\text{Cov}(\mathbf{u}_e, \mathbf{u}_{e'}) \\
&= \text{Cov}\left( \sum_{p:p\in e} \boldsymbol{\beta}_p \mathbf{1}_{ep}, \sum_{p':p'\in e'} \boldsymbol{\beta}_{p'} \mathbf{1}_{e'p'} \right) \\
&= \sum_{p\in e, p'\in e'} \boldsymbol{\beta}_p \boldsymbol{\beta}_{p'} \text{Cov}(\mathbf{1}_{ep}, \mathbf{1}_{e'p'}) \\
&\quad \because \text{Cov is bilinear} \\
&= \sum_{p\in e, e'} \boldsymbol{\beta}_p^2 \text{Cov}(\mathbf{1}_{ep}, \mathbf{1}_{e'p}) \\
&\quad \because \text{mutations at different } p \text{ are independent.}
\end{aligned}
\tag{A10}
$$

The last line's simplification follows from the independence of indicator variables $\mathbf{1}_{ep}$ across different positions. When $e$ and $e'$ have no overlapping positions, the summation is trivially zero as $\{p : p \in e, e'\}$ will be an empty set. It's also zero if the two edges are not ancestral to each other, as mutations running on separate edges are also independent. Otherwise:

$$
\begin{aligned}
&\text{Cov}(\mathbf{1}_{ep}, \mathbf{1}_{e'p}) \\
&= \mathbb{E}[\mathbf{1}_{ep}\mathbf{1}_{e'p}] - \mathbb{E}[\mathbf{1}_{ep}]\mathbb{E}[\mathbf{1}_{e'p}] \\
&\quad \because \text{Cov}(X, Y) = \mathbb{E}[XY] - \mathbb{E}[X]\mathbb{E}[Y] \\
&= \mathbb{E}[\mathbb{E}[\mathbf{1}_{ep}\mathbf{1}_{e'p} \mid \mathbf{1}_{ep}]] - l_e u_{ep} l_{e'} u_{e'p} \\
&\quad \because \mathbb{E}[X] = \mathbb{E}[\mathbb{E}[X \mid Y]] \\
&= \mathbb{E}[\mathbf{1}_{ep}\mathbb{E}[\mathbf{1}_{e'p} \mid \mathbf{1}_{ep}]] - l_e u_{ep} l_{e'} u_{e'p} \\
&\quad \because \mathbb{E}[XY \mid X] = X\mathbb{E}[Y \mid X] \\
&= \mathbb{P}(\mathbf{1}_{ep} = 0) \cdot 0 \cdot l_{e'} u_{e'p} \\
&\quad + \mathbb{P}(\mathbf{1}_{ep} = 1) \cdot 1 \cdot 0 - l_e u_{ep} l_{e'} u_{e'p} \\
&= -l_e u_{ep} l_{e'} u_{e'p}.
\end{aligned}
\tag{A11}
$$

This latter quantity is small because it is proportional to the per base-pair mutation rate squared. Not allowing child mutations gives $\mathbb{E}[\mathbf{1}_{e'p} \mid \mathbf{1}_{ep} = 1] = 0$, provided $e$ is an ancestor of $e'$ without the loss of generality. Therefore, we can justify the diagonal covariance matrices of random effects $\mathbf{u}$ by setting all terms involving $u_{ep}^2$ to zero.

In summary, we have the resulting generalization, that allows varying mutational variance along the genome and through time: The distribution of a complex trait follows a linear mixed model conditioned on the sample's ARG $\mathcal{T} = (\mathcal{N}, \mathcal{E})$:

$$
\mathbf{y} = \mathbf{u}_0 + \mathbf{Zu} + \boldsymbol{\varepsilon},
\tag{A12}
$$

with a constant vector $\mathbf{u}_0$ (all entries have the same value). The random effects $\mathbf{u}$'s covariance matrix is:

$$
\boldsymbol{\Sigma}_{\mathbf{u}} = \text{diag}[l_e s_e \tau_e^2]_{e=1,\dots,E},
\tag{A13}
$$

where $\tau_e^2 = \frac{1}{s_e} \sum_{p:p\in e} \boldsymbol{\beta}_p^2 u_{ep}$. The residual $\boldsymbol{\varepsilon}$ contains the contribution of non-genetic variables. We call the covariance matrix $\text{Var}(\mathbf{Zu}) = \mathbf{Z}\boldsymbol{\Sigma}_{\mathbf{u}}\mathbf{Z}^{\mathsf{T}}$ as the generalized branch GRM. In the model we use in the main paper, we further assume that $\boldsymbol{\beta}_p \sim \text{Normal}(\boldsymbol{\mu}_p, \tau^2)$. Then, $\text{Var}(\mathbf{Zu})$ further reduces to

$$
\text{Var}(\mathbf{Zu}) = \tau^2 \mathbf{B}
\tag{A14}
$$

in which $\mathbf{B}$ is the branch GRM from Lehmann et al. (2026). $\boldsymbol{\mu}_p$ does not affect the GRM and is absorbed to the intercept as in equation (A8) and (A9).

# Appendix B: Variance of $V_G$

First, note that if $X$ and $Y$ are Gaussian, then Isserlis' Theorem (Isserlis 1918) says that:

$$\mathbb{E}[X^2 Y^2] = \mathbb{E}[X^2]\mathbb{E}[Y^2] + 2\mathbb{E}[XY]^2.$$

Expanding out $\mathrm{Var}(V_G)$, we will use this fact with $X = (\mathbf{P_N g})_n$ and $Y = (\mathbf{P_N g})_m$. Continuing with the notation from equation (18):

$$
\begin{aligned}
&\mathrm{Var}(V_G) \\
&= \mathrm{Var}\!\left(\frac{1}{N}\sum_{n=1}^{N}(\mathbf{g}_n - \overline{\mathbf{g}})^2\right) \\
&= \frac{1}{N^2}\left(\mathbb{E}\!\left[(\mathbf{P_N g})^T(\mathbf{P_N g})(\mathbf{P_N g})^T(\mathbf{P_N g})\right] - \mathbb{E}\!\left[(\mathbf{P_N g})^T(\mathbf{P_N g})\right]^2\right) \\
&= \frac{1}{N^2}\sum_{n=1}^{N}\sum_{m=1}^{N}\left(\mathbb{E}\!\left[(\mathbf{P_N g})_n^2(\mathbf{P_N g})_m^2\right] - \mathbb{E}\!\left[(\mathbf{P_N g})_n^2\right]\mathbb{E}\!\left[(\mathbf{P_N g})_m^2\right]\right) \quad \text{(B1)}\\
&= \frac{1}{N^2}\sum_{n=1}^{N}\sum_{m=1}^{N}\mathbb{E}\!\left[(\mathbf{P_N g})_n(\mathbf{P_N g})_m\right]^2 \\
&= \tau^4\frac{1}{N^2}\sum_{n=1}^{N}\sum_{m=1}^{N}\left(\widetilde{\mathbf{B}}_{n,m}\right)^2.
\end{aligned}
$$

Since the offdiagonals of $\widetilde{\mathbf{B}}$ are $\mathcal{O}(1/N)$ (where recall that $N$ is the sample size), the last line tells us that $\mathrm{Var}(V_G) = \mathcal{O}(1/N)$, and hence, $\tau^2\mathrm{tr}(\widetilde{\mathbf{B}})/N$ is a consistent estimator of $V_G$, with an error that is of order $1/\sqrt{N}$.

# Appendix C: Correctness of Algorithm T

Here we present an algorithm for efficient simulation from the ARG-LMM model. This serves several purposes: first, the structure of the algorithm helps to highlight the dependency structures within the ARG. Second, the simplest tree-by-tree trait simulation algorithm for a large ARG could be computationally very costly, and this algorithm is much more efficient. Third, to illustrate how a key part of the matrix-vector product algorithm in Lehmann et al. (2026) works.

We begin with a precise explanation of what we'd like to simulate. Consider a given tree $T$ that extends over some span $s_T$ of the genome, and for an edge or sub-edge $e$ and a node $n$ write $e \geq_T n$ if $e$ is ancestral to $n$ in $T$. Each edge $e \in T$ connects a child $c(e)$ to a parent $p(e)$, and so has area $s_T(t_{p(e)} - t_{c(e)})$ in this tree. For each edge in each tree, let the edge effect $\mathbf{u}_{e,T}$ be a Gaussian random variable with mean zero and variance $s_T(t_{p(e)} - t_{c(e)})$. Then, the genetic value of a node $n$—the quantity we seek to simulate—is equal to $\sum_T \sum_{e \geq_T n} \mathbf{u}_{e,T}$. However, if the subtree below an edge $e$ does not change across trees $T_i, T_{i+1}, \ldots, T_j$, then these effects can be combined: $\mathbf{u}_{e,T_i} + \cdots + \mathbf{u}_{e,T_j}$ is inherited together by the same set of nodes. By additivity of Gaussian distributions, this combined effect is itself Gaussian with variance equal to the combined areas, and so we aim to simulate $\mathbf{Zu}$, where $\mathbf{Z}$ is the individual–sub-edge design matrix defined above and $\mathbf{u}$ is a vector of sub-edge effects from a Gaussian distribution with mean zero variance equal to sub-edge areas. (We emphasize that that in practice a single edge in the ARG can subtend many different subtrees, so that each $e$ denotes a sub-edge as defined above.)

The algorithm works, roughly, as follows. The "value" $v(n)$ of each node caches a total contribution to genetic value that is inherited by every node in the current subtree below $n$. The "current value" of node $n$ is obtained by summing up the values of all nodes above $n$ in the current tree—so, if the tree changes, we must make adjustments to maintain a correct current value. When an edge with parent $p$ and child $c$ is removed from the tree, two things happen. First, the effect of that edge is simulated, and added to $v(c)$. Then, the value of $p$ and every node above $p$ must be added to $v(c)$, because $c$ is no longer below those nodes. Adding an edge creates a different problem: after adding the edge, $c$ is below $p$ in the tree, but $v(p)$ contains effects from previous parts of the genome. Therefore, we subtract the values of $p$ and all nodes above $p$ from $v(c)$, so that these effects cancel out when summing up the tree from $c$. At the end of the algorithm all edges are removed, so we end up with the empty tree, and so $v(n)$ will be equal to the genetic value of node $n$.

**Algorithm T** (Trait genetic value simulation). Given a sequence of positions $0 = b_0 < b_1 < \cdots < b_K = L$ that are recombination breakpoints along the genome of length $L$, and corresponding sequences of edges to remove $(R_k)$ and add $(A_k)$ at each position, sample the genetic values $g_s$ for $1 \leq s \leq n_S$, as long as all samples are leaves in all trees. Let $T$ be the current tree, $\ell_T(n) = t_{p(n)} - t_n$ be the length of the edge above node $n$ in $T$ (or zero, if $n$ has no parent), initialize $k = 1$, $x(n) = 0$, and $v(n) = 0$ for all $n \in \mathcal{N}$. At all times, define $z(n) = \sqrt{\ell_{T_k}(n)(b_k - x(n))}$. The $k$th tree $T_k$ has span $s_{T_k} = b_k - b_{k-1}$.

> **T1.** [Remove edges] For each edge $(c, p) \in R_k$, and for each node $n \geq_T p$, sample $\mathbf{w}_n \sim \mathrm{Gaussian}(0, 1)$; set $v(n) \mathrel{+}= z(n)\mathbf{w}_n$; then set $v(c) \mathrel{+}= v(n)$ and $x(n) = b_k$. Then, set $x(c) = b_k$ and remove the edge.
>
> **T2.** [Add edges] For each edge $(c, p) \in R_k$, and for each node $n \geq_T p$, sample $\mathbf{w}_n \sim \mathrm{Gaussian}(0, 1)$; set $v(n) \mathrel{+}= z(n)\mathbf{w}_n$; then set $v(c) \mathrel{-}= v(n)$ and $x(n) = b_k$. Then, set $x(c) = b_k$ and add the edge.
>
> **T3.** [Iteration] If $k < K$, set $k = k + 1$ and return to **T1**. Otherwise, set $g_s = v(s)$ for $1 \leq s \leq n_S$ and finish.

The algorithm proceeds by storing and updating intermediate quantities $v$, $z$, and $x$ indexed by nodes. The genetic value is recovered at the end of the algorithm by aggregating these intermediate values efficiently. Updating the intermediate values is also efficient because it only alters the entries that are ancestral to the added or deleted edge in the local tree while leaving all other entries constant.

To prove that the algorithm works, we first reformulate the algorithm so that we have the edge effects beforehand, making the algorithm deterministic. To this end, for each node $n$ in the $k$th tree, let $\mathbf{z}_{n,k}$ have a Gaussian distribution with mean zero and variance equal to the span of the tree, $b_k - b_{k-1}$. Recall that $x(n)$ is the last position at which node $n$ was visited, and so we can define $k(n)$ so that $b_{k(n)} = x(n)$. Now, define $Z(n) = \sqrt{\ell_{T_k}(n)}(\mathbf{z}_{n,k(n)} + \cdots + \mathbf{z}_{n,k})$. Comparing to Algorithm T, $Z(n)$ and $z(n)\mathbf{w}_n$ have the same distribution; the only difference is that we sample from the Gaussian here in several steps. Thus, where steps **T1** and **T2** say "set $v(n) \mathrel{+}= z(n)\mathbf{w}_n$," instead replace these with the equivalent instruction "set $v(n) \mathrel{+}= Z(n)$."

To show that the algorithm calculates the genetic value, we first show how to obtain the genetic value for each node based on the genome up until the current position. Then, we show that steps **T1** and **T2** in Algorithm T do not change the genetic value.

First, define the "working genetic value," which is the genetic value of node $n$ calculated only the genome up until position $b_k$, as

$$g_k(n) = \sum_{h=1}^{k} \sum_{u \geq_T n} \sqrt{\ell_{T_h}(u)} \mathbf{z}_{u,h}.$$

This has the correct distribution because $\sqrt{\ell_{T_h}(u)} \mathbf{z}_{u,h}$ has a Gaussian distribution with mean zero and variance equal to $(b_h - b_{h-1})\ell_{T_h}(u)$, which is the area of the edge above $u$ in tree $h$. By additivity of Gaussians, the contribution of a given edge that spans multiple trees is again Gaussian has mean zero and variance equal to its area.

Also, define

$$\hat{g}(n) = \sum_{r : r \geq_T n} (v(r) + Z(r)). \tag{C1}$$

We claim that at any point in the algorithm,

$$g_k(n) = \hat{g}(n), \quad \text{for all } n. \tag{C2}$$

At the end of the algorithm, the value returned is $\hat{g}(s)$, because tree is empty at termination, so the right-hand side reduces to $v(s)$. $z(s)$ is zero because $x(s) = b_K$ (recall that $z(n) = \sqrt{\ell_{T_k}(n)(b_k - x(n))}$). Therefore, if equality (C2) holds throughout the algorithm, then $v(s)$ is the desired genetic value, $g_K(s)$.

We prove equation (C2) by induction. The equality trivially holds at the beginning of the algorithm, because both sides are zero. Next we prove that each of the three steps in the algorithm preserves the equality (C2).

**Step T3** is the easiest to analyze. By the induction step, we assume that we began this step with $\hat{g}(n) = g_k(n)$. The current tree $T$ and the intermediate quantities $v$ and $x$ do not change. Advancing the position from $k$ to $k' = k + 1$ changes $Z(n)$ by adding $\sqrt{\ell_{T_k}(n)} \mathbf{z}_{n,k+1}$, and so changes $\hat{g}(n)$ by

$$\sum_{r : r \geq_T n} \sqrt{\ell_{T_k}(r)} \mathbf{z}_{r,k+1}.$$

Comparing to equation (C1), this is equal to $g_{k+1}(n) - g_k(n)$, and so at the end of this step, $\hat{g}(n) = g_{k'}(n)$, as desired.

This is the only step where the working genetic value changes. In the other two steps, $v$, $z$, and $x$ change, but $\hat{g}(n)$ remains constant, thus preserving equation (C2). We prove this only for **T2**; the argument is basically the same for **T1**.

Suppose that we added edge $(c, p)$ where $c$ and $p$ are child and parent nodes of the edge, respectively. Call the tree at the start of the step $T$, and the tree after the step $T'$, and similarly for $x'$, $v'$, $Z'$, and $\hat{g}'$. Since adding an edge only increases the set of nodes above $n$, the difference in $\hat{g}(n)$ caused by **T2** is

$$\begin{aligned}
&\hat{g}'(n) - \hat{g}(n) \\
&= \sum_{r : n \leq_{T'} r} (v'(r) + Z'(r)) - \sum_{r : n \leq_T r} (v(r) + Z(r)) \\
&= \sum_{r : n \leq_{T'} r, n \, / \leq_T r} (v'(r) + Z'(r)) \\
&\quad + \sum_{r : n \leq_T r} (v'(r) + Z'(r) - v(r) - Z(r)).
\end{aligned} \tag{C3}$$

We can divide the nodes of $\mathcal{N}$ into four categories as (1) the child node $c$, (2) nodes below $c$, (3) the parent node $p$ and nodes above $p$, and (4) all other nodes. Since $v(r)$ and $Z(r)$ only change for nodes that are above $c$ in $T'$, evidently $\hat{g}'(n) = \hat{g}(n)$ for nodes in the last category. Next, note that $v(n) + Z(n) = v'(n) + Z'(n)$ for nodes $n$ that are above $p$: this is because **T2** first sets $v'(n) = v(n) + Z(n)$, then sets $x(n) = b_k$, which has the effect that $Z'(n) = 0$. Therefore, $\hat{g}'(n) = \hat{g}(n)$ for nodes $n \geq_T p$.

This leaves only nodes $n \leq_{T'} c$. The first term in equation (C3) sums over $\{r : n \leq_{T'} r, n \, / \leq_T r\}$: nodes that are above $n$ after adding the edge, but were not above $n$ before the addition. Hence, for nodes below $c$, this set is simply all nodes above $p$, and the first summation is

$$\sum_{r : p \leq_{T'} r} (v'(r) + Z'(r)). \tag{C4}$$

The only nodes that change $x$ are those above $p$, so for nodes $r \leq_T c$, $Z'(r) = Z(r)$. Furthermore, $v'(r) = v(r)$ for nodes strictly below $c$. Thus, the summand of the second summation is $v'(r) + Z'(r) - v(r) - Z(r)$ is equal to 0, except for $r = c$, in which case it is $v'(c) - v(c)$. Therefore, the second summation is just $v'(c) - v(c)$. Now, $v(c)$ changes because this step, for each $r \geq_T p$, first sets $v'(r) = v(r) + Z(r)$, and then subtracts $v'(r)$ from $v(c)$. As a result, still for nodes $n$ below $c$, the second summation is

$$v'(c) - v(c) = -\sum_{r : p \leq_T r} (v(r) + Z(r)). \tag{C5}$$

Putting these together, the two summations in equation (C3) cancel, so $\hat{g}'(n) = \hat{g}(n)$ for all remaining nodes.

In summary, **T1** and **T2** modify the intermediate values $v$, $x$, and $Z$ while leaving the working genetic value constant so that **T3** correctly updates the genetic value. See **Efficient ARG-LMM simulation** for more details of fully simulating the trait based on this algorithm.

## Appendix D: Submatrix computation

`tskit` provides multiplication by $\mathbf{B}$ (and $\mathbf{B}^{-1}$ via conjugate gradient), and for its submatrices, one can multiply an incidence matrix before and after multiplying $\mathbf{B}$ to perform matrix-vector multiplication. For instance, $\mathbf{B}_{n,o}\mathbf{v}$ can be done by:

$$\mathbf{B}_{n,o}\mathbf{v} = \mathbf{A}_n \cdot \mathbf{B} \cdot \mathbf{A}_o^T \mathbf{v}, \tag{D1}$$

where $\mathbf{A}_n$ and $\mathbf{A}_o$ are incidence matrices with each row corresponding to the non-phenotyped and observed individuals. For each row, only one column that corresponds to the index of the row in the full list of individuals are set to 1 and 0 otherwise. Multiplication by incidence matrices $\mathbf{A}_{\bullet}$ can be done very quickly as they are very sparse.

The conditional variances of the predicted genetic values may be useful as a measure of precision for individual BLUPs, but in large-scale applications assembling the full covariance matrix (equation 20) is infeasible. However, we can obtain unbiased estimates of the diagonal using the XDiag algorithm in Epperly et al. (2024), which only requires matrix-vector products of equation (20) against a fixed number of random test vectors.

*Editor: N. Zaitlen*