## [Peer Review File · Genetics]

Genetic prediction with ARG-powered linear algebra

Hanbin Lee, Nathaniel Pope, Jerome Kelleher, Gregor Gorjanc, and Peter Ralph

NOTE: The reviews and decision letters are unedited and appear as submitted by the reviewers.

In extremely rare instances and as determined by a Senior Editor or the EIC, portions of a review may be redacted. If a review is signed, the reviewer has agreed to no longer remain anonymous.

The review history appears in chronological order.

Review Timeline:

Submission Date:	2025-11-29
Editorial Decision:	2026-01-21
Revision Received:	2026-02-05
Accepted:	2026-03-12

January 21, 2026

RE: GENETICS-2025-308825

Dear Dr. Lee:

I am pleased to accept your manuscript titled "Genetic prediction with ARG-powered linear algebra" for publication in GENETICS, pending minor revision.

Please submit your revision along with a brief description of how you modified the manuscript in response to the reviewers' concerns and suggestions (which can be viewed at the bottom of this email. Most important are comparisons to SNP based GRM methods and a more comprehensive discussion of the tradeoffs between the two. I expect you should be able to submit a revised manuscript within 30 days. A suitably revised manuscript will be acceptable for publication; I don't expect to send it out for review.

When revising the ms., please make an effort to shorten it, because that almost always improves a manuscript. We urge authors to heed the advice of Strunk and White: "omit needless words"¹. Follow this link to submit the revised manuscript: Link Not Available

Thank you for submitting this story to Genetics.

Sincerely,

Noah Zaitlen
Associate Editor
GENETICS

Approved by:
Hongyu Zhao
Senior Editor
GENETICS

Reviewer comments:

Reviewer #1 :

Lee et al present an ARG-LMM for inference of variance components and phenotype prediction using ARGs. Crucially they are able to use ARG-based matrix-vector multiplication and modern linear algebra techniques to develop a fast REML algorithm that they demonstrate is significantly better than Haseman-Elston approaches. They also demonstrate that the ARG-BLUP is pretty accurate. There's some additional theoretical discussion of the generative model and connection to various models existing in the literature.

On a technical level, this paper is hard to fault. I think it is very thorough and careful and the methods proposed are valid to the best of my ability to assess them. Combined with the manuscript's exploration of connections of the ARG-LMM to classical methods based on pedigrees and realized GRMs, I think that this paper is already interesting and worthy of publication.

The only major addition I would like to see in the manuscript, and I do not think it is necessary, is a more explicit comparison to using the SNP-GRM for the same approaches. The authors bring up that it's slightly odd to "ignore" the SNPs when using the ARG-GRM, but state that in some cases (e.g. sparse SNP datasets) there is evidence that it outperforms the SNP-GRM. I'm probably fine with the authors leaving it at that, and referencing other papers, but it could make this paper stronger and more self-contained to make those comparisons explicitly. Similarly, the authors could compare running time of fast methods for variance component inference from the SNP-GRM (e.g. BOLT-LMM) to their approach. Again, I do not think that this is an essential addition to the manuscript, but it would merely be nice.

Otherwise, I only have some minor comments

- 1) In the paragraph starting line 120, especially due to the reference to phylogenetic models of Brownian motion, it might be appropriate to cite Schraiber et al (2024).
- 2) Line 152: I think it's probably worth emphasizing here that this is (implicitly) a neutral model. The authors highlight this later, but I think being up front about it is useful here, too.
- 3) In some presentations of the ARG-GRM, it's "standardized" by something that's the ARG-equivalent of $p(1-p)$, e.g. in Fan et al's approach. The authors do bring this up in the paragraph on 591, but I think it might be worth bringing up earlier (e.g. the definition of B below line 164). It'd also be useful to know if this kind of standardization can be done in their framework (I think so?)
- 4) In the paragraph starting line 182, I think a citation to Schraiber and Landis (2015) and Koch (2019) would be appropriate, as they discuss a similar neutral model in detail (while averaging over the genealogy).
- 5) The authors point out the trace cancelation in the calculation of the average information, as if it is a lucky thing in their specific case, but my memory of how AI works is that the trace cancelation is the whole point and happens generally? I assume the authors know this and this is just a quirk of my reading of their writing, but it might be nice to clarify (or else point out that I am mistaken).
- 6) Line 313: "criteria" -> "criterion"
- 7) Figure 4: I think that making the green line actually a different line type than the other lines would be useful to indicate that it is fundamentally different from the blue and orange lines, and represents a maximum possible accuracy, would help the readability of the figure.
- 8) In the "more realistic simulations" with a spatial model, the spatial model is only active for 150 time units, before being recapitated. I would think that means that almost all the time in the coalescent tree is in the recapitation, which very little signal due to the spatial model. Is that accurate?

I prefer to sign my reviews. My name is Joshua Schraiber.

References:

- Schraiber, J. G., Edge, M. D., & Pennell, M. (2024). Unifying approaches from statistical genetics and phylogenetics for mapping phenotypes in structured populations. *PLoS biology*, 22(10), e3002847.
- Fan, C., Mancuso, N., & Chiang, C. W. (2022). A genealogical estimate of genetic relationships. *The American Journal of Human Genetics*, 109(5), 812-824.
- Schraiber, J. G., & Landis, M. J. (2015). Sensitivity of quantitative traits to mutational effects and number of loci. *Theoretical population biology*, 102, 85-93.
- Koch, E. M. (2019). The effects of demography and genetics on the neutral distribution of quantitative traits. *Genetics*, 211(4), 1371-1394.

Reviewer #2 :

This paper proposes a computational model of fitting linear mixed models using ancestral recombination graphs (ARGs) as the genetic covariance structure to enable scaleable variance component estimation and genomic prediction in large cohorts. The authors claim how ARG-based covariance operators can be used without explicitly forming dense matrices by applying succinct tree-sequence representations and randomized linear algebra.

The paper provides a clear formulation showing how ARG-derived covariance structures can be embedded into linear mixed models, and achieves near-linear scaling by using tree sequences. The paper includes numerical comparisons between true ARGs, inferred ARGs, and GRM-based methods (genomic relationship matrix), helping to illustrate the performance difference.

I'd like to list several points where clarification, justification, strengthening could improve the paper

1. From a linear mixed model perspective, both ARGs and GRMs enter the model as covariance structures; and under standard additive assumptions and sufficiently dense markers, a GRM can be viewed as an empirical approximation to the covariance implied by the underlying ARG. ARG is a different parameterization of relatedness rather than providing fundamentally new

inferential power. The paper could benefit from explicitly discussing when and why ARG-based covariance is expected to theoretically outperform GRMs, and when the two are expected to yield nearly identical prediction performance.

2. The authors state that ARGs provide a "time axis" that leads to more interpretable results (L658). It may be true in principle, it is not clear how this advantage is actually exploited in the current model. The model does not incorporate time-related effects explicitly. It reads more like the time axis is a latent property of the data structure.

3. The paper relies heavily on randomized Haseman-Elston (HE) regression for variance component estimation, but may not explain sufficiently how this choice is motivated.

4. The paper demonstrates that ARG-based BLUP is feasible, but it is less clear that it provides meaningful advantages for genomic prediction accuracy. The strong advantage of ARGs may be in variance decomposition and interpretability.

Reviewer #3 :

Major comments:

1. As inferred ARGs will necessarily be used in real data practices, I would like to see some more discussions on the comparison between ARG inference methods. For example, Relate could be run for 2000 sequences, and ARG-Needle/Threads could be run for more. It would be useful for the authors to compare the performance between these methods and give general guidelines to the potential users;

2. Another aspect which is generally missing from the article is that the benchmark has not compared the performance using whole genome sequences versus ARGs (true and inferred). The benchmark as of now focuses on comparing the performance between inferred and true ARGs and makes the point that the loss is not significant. While that seems to be true, there is one fundamental question which remains unanswered: does ARG-based inference provide additional benefit other than computational efficiency? If the ARG-based computation only saves a tolerable runtime but results in worse accuracy, I hardly think people will use it. I would like the authors to make the comparison between using whole genomes and inferred ARGs, to address the above concern;

3. There is something non-intuitive in Figure 3 and 4. In Figure 4, the authors show that the sample correlation performance using inferred ARGs is getting closer with using true ARGs. However, in Figure 3, the MSE gap (blue vs green) seems to get larger with large sample sizes. Can the author explain this obvious inconsistency?

4. Since the method relies on building ARGs from data, it would also be good to discuss the difference between using genotype array data and whole genome sequences, as they hugely impact the ARG inference accuracy. It would also be nice to discuss the robustness of ARG-based estimator and genotype-based estimator in these 2 cases;

5. It would be nice to say something about the memory use of their method at different data sizes. I am sure that is a benefit of the method.

Associate Editor comments:

Reviewer 1:

(1.1) The only major addition I would like to see in the manuscript, and I do not think it is necessary, is a more explicit comparison to using the SNP-GRM for the same approaches. The authors bring up that it's slightly odd to "ignore" the SNPs when using the ARG-GRM, but state that in some cases (e.g. sparse SNP datasets) there is evidence that it outperforms the SNP-GRM. I'm probably fine with the authors leaving it at that, and referencing other papers, but it could make this paper stronger and more self-contained to make those comparisons explicitly.

Reply: We added a SNP genotype GRM to the comparison with the definition following the standard formula:

$$\text{GRM}_{ij} = \frac{1}{M} \sum_{k=1}^M \frac{(x_{ik} - 2p_k)(x_{jk} - 2p_k)}{2p_k(1 - p_k)}$$

We filtered out variants with sample frequency lower than 0.005 to avoid large denominators. The result is presented in Figure 7, and described at (p. 22, l. 574).

Like with pedigree and Monte-Carlo ARG-GRMs, we fitted the variance component using a custom `nloptr`-based script. The algorithm is from Meyer (1985) and Lippert et al. (2011), where we rotate the trait and the covariates by the eigenvectors of the GRM.

(1.2) Similarly, the authors could compare running time of fast methods for variance component inference from the SNP-GRM (e.g. BOLT-LMM) to their approach. Again, I do not think that this is an essential addition to the manuscript, but it would merely be nice.

Reply: We added runtime comparison between `tslmm` and BOLT-LMM. The comparison shows that runtime of `tslmm` scales more gradually than that of BOLT-LMM. See Figure 4 and (p. 14, l. 408). However, we've been careful not to present this as a careful comparison with modern genotype-based methods; doing justice to that work falls to another paper.

(1.3) In the paragraph starting (p. 4, l. 125), especially due to the reference to phylogenetic models of Brownian motion, it might be appropriate to cite Schraiber, Edge, and Pennell (2024).

Reply: Good suggestion; we have cited this paper. (p. 4, l. 125)

(1.4) (p. 3, l. 116) I think it's probably worth emphasizing here that this is (implicitly) a neutral model. The authors highlight this later, but I think being up front about it is useful here, too.

Reply: We added a sentence to the paper to mention this. (p. 3, l. 116)

(1.5) In some presentations of the ARG-GRM, it's "standardized" by something that's the ARG-equivalent of $p(1-p)$, e.g. in Fan et al's approach. The authors do bring this up in the paragraph at (p. 24, l. 655), but I think it might be worth bringing up earlier (e.g. the definition of B at (p. 5, l. 170)). It'd also be useful to know if this kind of standardization can be done in their framework (I think so?)

Reply: Good idea; the short answer to this is that yes, we can incorporate frequency normalization, but have not, essentially because this normalization doesn't stem from a generative model. We have added a note about this at (p. 5, l. 171).

(1.6) *In the paragraph starting (p. 6, l. 194), I think a citation to Schraiber and Landis, 2015 and Koch, 2019 would be appropriate, as they discuss a similar neutral model in detail (while averaging over the genealogy).*

Reply: Again, good ideas; this is done. (p. 6, l. 194)

(1.7) *The authors point out the trace cancellation in the calculation of the average information, as if it is a lucky thing in their specific case, but my memory of how AI works is that the trace cancellation is the whole point and happens generally? I assume the authors know this and this is just a quirk of my reading of their writing, but it might be nice to clarify (or else point out that I am mistaken).*

Reply: This is true. The whole point of AI-REML is the trace cancellation which holds whenever the total covariance matrix is a linear combination of parameters. Interestingly, although not directly related to the paper, empirical evidence and heuristics suggest that AI-REML still works even if the linearity of the covariance matrix doesn't hold. See Zhu and Wathen (2018). We updated the text to reflect this (p. 11, l. 308).

(1.8) *(p. 12, l. 343) "criteria" -> "criterion"*

Reply: Thank you for spotting this error. We've corrected it. (p. 12, l. 343)

(1.9) Figure 6 *I think that making the green line actually a different line type than the other lines would be useful to indicate that it is fundamentally different from the blue and orange lines, and represents a maximum possible accuracy, would help the readability of the figure.*

Reply: We changed it to a dotted black line (Figure 6).

(1.10) *(p. 20, l. 523) In the "more realistic simulations" with a spatial model, the spatial model is only active for 150 time units, before being recapitated. I would think that means that almost all the time in the coalescent tree is in the recapitation, which very little signal due to the spatial model. Is that accurate?*

Reply: Good question; one answer to your question is to note that in Figure 7A, we can see lots of local correlation in trait value, which indicates shared ancestry relevant to trait prediction. It is true that most "coalescent time" is before this, but nonetheless we expect that the top few PCs to reflect geography after this period, and there to be potentially more smaller spatial structure (i.e., clusters of close relatives). This was the main motivation for the experiment: since having "different" eigenstructure for the covariance matrix can lead to different convergence properties for the randomized linear algebra methods we're using. We have added an additional few sentences that point this out more explicitly (p. 21, l. 560).

Reviewer 2:

(2.1) *From a linear mixed model perspective, both ARGs and GRMs enter the model as covariance structures; and under standard additive assumptions and sufficiently dense markers, a GRM can be viewed as an empirical approximation to the covariance implied by the underlying ARG. ARG is a different parameterization of relatedness rather than providing fundamentally new inferential power. The paper could benefit from*

explicitly discussing when and why ARG-based covariance is expected to theoretically outperform GRMs, and when the two are expected to yield nearly identical prediction performance.

Reply: We agree that this is an important point and common question readers will have. We have updated our Discussion to (hopefully) more directly address this point (p. 24, l. 664). In particular, we've tried to make the points that Zhu et al. (2025) has obtained good empirical results in this direction (that ARG-based GRMs can add additional power in some circumstances), but more empirical work is needed.

(2.2) (p. 26, l. 741) The authors state that ARGs provide a "time axis" that leads to more interpretable results (p. 26, l. 741). It may be true in principle, it is not clear how this advantage is actually exploited in the current model. The model does not incorporate time-related effects explicitly. It reads more like the time axis is a latent property of the data structure.

Reply: We agree that our comment wasn't clear enough. Now, we have reminded the reader that the time axis is useful in this paper because it is reflected in the units of τ^2 (p. 26, l. 741).

(2.3) The paper relies heavily on randomized Haseman-Elston (HE) regression for variance component estimation, but may not explain sufficiently how this choice is motivated.

Reply: We've added a sentence giving the motivation for using this as a starting location at (p. 12, l. 358). (We don't think that our use of it is particularly crucial, however; also see discussion of HE in the Introduction, around (p. 3, l. 82).)

(2.4) The paper demonstrates that ARG-based BLUP is feasible, but it is less clear that it provides meaningful advantages for genomic prediction accuracy. The strong advantage of ARGs may be in variance decomposition and interpretability.

Reply: We agree generally, but also think that there may be advantages for accuracy in some situations (e.g., when genotyping data is incomplete, as observed for GWAS by Zhu et al., 2025). We have mentioned this in the Discussion (see point above and section at (p. 24, l. 664)). An explicit comparison to the usual SNP-GRM (genotype GRM) was also added to Figure 7.

Reviewer 3:

(3.1) As inferred ARGs will necessarily be used in real data practices, I would like to see some more discussions on the comparison between ARG inference methods. For example, Relate could be run for 2000 sequences, and ARG-Needle/Threads could be run for more. It would be useful for the authors to compare the performance between these methods and given general guidelines to the potential users;

Reply: This would be an excellent thing to explore, but we believe it is beyond the scope of this paper. For instance, the strengths and weaknesses of different methods are expected to be different at different sample sizes and in different situations (data quality; structure; etcetera), so that an informative exploration would make this paper far too long. There's an ongoing effort to measure how sensitive ARG-based methods are to ARG inference methods, e.g. Peng, Mulder, and Edge (2025). We have added some more discussion of this point to (p. 24, l. 685).

(3.2) Another aspect which is generally missing from the article is that the benchmark has not compared the performance using whole genome sequences versus ARGs (true and inferred). The benchmark as of now focuses on compare the performance between inferred and true ARGs and make the point that the loss is not significant. While that seems to be true, there is one fundamental question which remains unanswered: does ARG-based inference provide additional benefit other than computational efficiency? If the ARG-based computation only saves a tolerable runtime but results in worse accuracy, I hardly think people will use it. I would like the authors to make the comparison between using whole genomes and inferred ARGs, to address the above concern;

Reply: We agree – the question of how and when it is beneficial to use ARGs is a significant open question in the field (and which needs some empirical work, in our opinion). To give a bit more information about this, we have added a comparison (see Figure 7) to the usual SNP-GRM (the genotype GRM), showing that we do not have worse results, even with inferred ARGs. The SNP set used to compute the genotype GRM contains all true causal variants alongside non-causal ones, with filtering to avoid large denominators in constructing the GRM. Furthermore, we now discuss in more detail around (p. 24, l. 664), other situations in which ARG-based methods might provide an advantage to SNP-based methods.

(3.3) There is something non-intuitive in Figure 5 and 6. In Figure 6, the authors shows that the sample correlation performance using inferred ARGs is getting closer with using true ARGs. However, in figure 5, the MSE gap (blue vs green) seems to get larger with large sample sizes. Can the author explain this obvious inconsistency?

Reply: This is an interesting observation, but after thinking hard about it, we don't think we can confidently say anything useful about this pattern. There are only two sample sizes (2,000 and 50,000) in Figure 5, and the latter sample size is below the largest sample size in Figure 6; and furthermore although the gap seen in Figure 5 does not get smaller, both blue and green bars in that figure do get smaller. It would be interesting to understand more, but the point of this paper is to describe the method, not to benchmark the behavior in different ARG inference situations.

(3.4) Since the method relies on building ARGs from data, it would also be good to discuss the difference between using genotype array data and whole genome sequences, as they hugely impact the ARG inference accuracy. It would also be nice to discuss the robustness of ARG-based estimator and genotype-based estimator in these 2 cases;

Reply: Again, we definitely agree, and would love to explore this, but we can't get too deep on this here, since it would be very speculative. We do point to the work by Zhu et al., 2025 in a few more places, who more directly study this exact question. We now refer more directly to this point at (p. 24, l. 685) and around (p. 24, l. 664).

(3.5) It would be nice to say something about the memory use of their method at different data sizes. I am sure that is a benefit of the method.

Reply: The memory scaling is given by the product of the number of nodes and test vectors used in trace estimation. We have updated the text at (p. 14, l. 413).

March 12, 2026

RE: GENETICS-2025-308825R1

Rev. Hanbin Lee
University of Michigan
Department of Statistics
1085 S. University Ave.
Ann Arbor, Michigan 48109

Dear Dr. Lee:

Congratulations, your manuscript titled "Genetic prediction with ARG-powered linear algebra" is accepted for publication in GENETICS! Many thanks for submitting your research to the journal.

To Proceed to Publication:

1. Format your article according to GENETICS style: <https://academic.oup.com/genetics/pages/author-guidelines>
2. Ensure that you comply with data and community resource citation guidelines:
<https://academic.oup.com/genetics/pages/author-guidelines#section-5-9-2>
3. Upload your final files at <https://genetics.msubmit.net>
4. Add oupsupport@scipris.com and genetics.oup@novatechset.com (or the domains @scipris.com and @novatechset.com) to your email program's "safe senders" list. You will be contacted by both at various points during the production process.

Notes:

- Your currently-accepted manuscript (unedited, as submitted, reviewed, and accepted) will be published at GENETICS and deposited into PubMed as an Advance Access article. Notify sourcefiles@thegsajournals.org before signing your license if you do not wish to publish your article via Advance Access.
- We invite you to submit an original color figure related to your paper for consideration as cover art. Please email your submission to the editorial office or upload it with your final files. You can submit a small-sized image for evaluation, and if selected, the final image must be a TIFF file 2513px wide by 3263px high (8.375 by 10.875 inches; resolution of 600ppi). Please avoid graphs and small type.
- After files are sent to Oxford University Press we use SciPris to manage article licensing and payment. If you do not have a SciPris account, you will receive an email from no-reply@scipris.com to sign up to use Oxford University Press' author portal. After logging in, follow the online instructions to sign your license and arrange any payment due.

If you have any questions or encounter any problems while uploading your accepted manuscript files, please email the editorial office at sourcefiles@thegsajournals.org.

Sincerely,

Noah Zaitlen
Associate Editor
GENETICS

Approved by:
Hongyu Zhao
Senior Editor
GENETICS

Review comments (if applicable):

Reviewer #1 :

The authors did a great job responding to comments.

Reviewer #3 :

I appreciate the authors' efforts to address my previous comments.